# Multiple processes of vocal sensory-motor interaction in primate auditory cortex

Joji Tsunada[1,2], Xiaoqin Wang ®[3] & Steven J. Eliades ®[1,4] ✉

Sensory-motor interactions in the auditory system play an important role in vocal self-monitoring and control. These result from top-down corollary discharges, relaying predictions about vocal timing and acoustics. Recent evidence suggests such signals may be two distinct processes, one suppressing neural activity during vocalization and another enhancing sensitivity to sensory feedback, rather than a single mechanism. Single-neuron recordings have been unable to disambiguate due to overlap of motor signals with sensory inputs. Here, we sought to disentangle these processes in marmoset auditory cortex during production of multi-phrased 'twitter' vocalizations. Temporal responses revealed two timescales of vocal suppression: temporally-precise phasic suppression during phrases and sustained tonic suppression. Both components were present within individual neurons, however, phasic suppression presented broadly regardless of frequency tuning (gating), while tonic was selective for vocal frequencies and feedback (prediction). This suggests that auditory cortex is modulated by concurrent corollary discharges during vocalization, with different computational mechanisms.

Vocal communication is a fundamental behavior shared by both humans and many animal species. Accurate communication requires auditory self-monitoring to detect and correct production errors[1], but the underlying neural processes that allow this self-monitoring remain uncertain. During vocal production, there is well-described suppression of neural activity in the auditory cortex that has been seen for both human speech[2–8] and non-human primate vocalization[9–12]. Simultaneously, the auditory cortex also exhibits a sensitivity to perturbations in auditory (sensory) feedback during vocal production that is enhanced compared to similar manipulations during passive listening[13–15]. This increased sensitivity appears to play an important role in feedback-dependent vocal control behaviors in both humans and animals[7,16,17]. Because these two processes, suppression and feedback sensitivity, are specific to vocal production and overlap in time, we have previously assumed that the two resulted from a single neural mechanism or process, particularly in primates where individual

neurons were found to exhibit a strong correlation between suppression, feedback sensitivity, and vocal compensation behaviors[13,16].

Recent evidence has emerged, however, that calls into question whether observed vocal suppression and feedback sensitivity share a common mechanism, or could represent independent processes. Intracranial neural recordings of high gamma activity over the human lateral temporal cortex, for example, exhibit both speech-induced suppression as well as feedback enhancement, but the two are often strongest at different cortical sites[7]. Other work has suggested that both general suppression and specific predictions about upcoming vocal production can co-exist during the preparation for speech[18]. In contrast, human intracranial recordings over Heschl's gyrus, where the primary auditory cortex is located, have found significant correlations between speech-related activity and feedback enhancement, particularly for measures of neural activity such as the frequency-following response[17]. Because these human experimental methods sample the

[1]Auditory and Communication Systems Laboratory, Department of Otorhinolaryngology: Head and Neck Surgery, University of Pennsylvania Perelman School of Medicine, Philadelphia, PA, USA. [2]Chinese Institute for Brain Research, Beijing, China. [3]Laboratory of Auditory Neurophysiology, Department of Biomedical Engineering, Johns Hopkins University School of Medicine, Baltimore, MD, USA. [4]Department of Head and Neck Surgery & Communication Sciences, Duke University School of Medicine, Durham, NC, USA. ✉e-mail: steven.eliades@duke.edu

average activity of larger numbers of neurons, it is unclear whether multiple sensory-motor processes are acting on individual neurons during vocalization, if they can be separated out, and what their properties might be.

Vocalization-induced modulation has been suggested to result from sensory-motor integration of vocal feedback and top-down signals originating in structures involved in the initiation and control of vocal production[19,20]. These motor signals, termed corollary discharges[21] or efference copies[22] have been implicated in a variety of sensory-motor processes spanning multiple sensory domains and across the animal kingdom[8,23–26]. Controversy exists, however, as to the information content and function of such pathways. Many corollary discharges function as non-specific inhibitory inputs from motor areas, the function of which can be to gate, block, or attenuate self-generated sensory inputs in a temporally-precise fashion[27,28]. In contrast, predictive corollary discharges have also been described, in which more-specific predictions about the expected sensory consequences of an action are relayed from motor areas and then compared to resulting sensory feedback[25,26]. Such predictions are integral to models of the sensory-motor control of movement[24,29].

It remains uncertain, however, as to the potential existence and role of different corollary discharges at the level of cortical neurons during vocal production. We, therefore, sought to determine the contribution these two putative processes, one gating and suppressive and one predictive, might be playing during vocalization-related modulation of individual neurons in the auditory cortex of marmoset monkeys (*Callithrix jacchus*). Because of the previously-observed correlations between vocal suppression and feedback sensitivity[13,16], the possible temporal concurrence of the two processes, and confounds by overlapping of sensory feedback with motor signals[30], we focused on the marmoset 'twitter' call rather than the long duration vocalizations previously studied. Twitters are vocalizations that consists of multiple short phrases interspersed with quiet intervals and, therefore, allow the potential to discriminate the effects of corollary discharges operating at different timescales, including individual phrases and over the whole vocal duration, and potentially disambiguate corollary discharge effects operating on multiple timescales as well as their interactions with different acoustic inputs. We found distinct phasic (phrase-specific) and tonic (whole vocalization) suppression of auditory cortex neurons during twitter call production, manifested as differences in activity between twitter phrases and inter-phrase intervals. To understand the auditory contributions and functions of these processes, we further compared responses during twitter production to those during passive playback of recorded vocalizations and frequency tuning, as well as responses during twitters perturbed with frequency-shifted vocal feedback. While many neurons exhibited only one of these tonic or phasic responses, others exhibited elements of both, suggesting the simultaneous presence of multiple processes of vocalization-induced suppression in the auditory cortex.

## Results

### Temporal patterns of vocal responses

We recorded activities in 3285 units from the auditory cortices of five marmoset monkeys while the animals made voluntary, self-initiated vocalizations, focusing on the responses during the multi-phrased twitter calls (Fig. 1a). Figure 1 (b–e) illustrates a sample unit's responses during twitter production. Although the neural activity suggested a phasic, and somewhat excitatory, response during vocal production (Fig. 1b, c: blue), the considerable variability in the duration of twitter phrases and inter-phrase intervals distorted the temporal response patterns when examining onset-aligned activities (Fig. 1c). This was particularly evident in later phrases, where accumulated variability in earlier phrase durations resulted in large overlaps of phrases and intervals of different calls and blurring of their respective neural

responses (Fig. 1c and Supplementary Fig. 1). Directly comparing activity during twitter phrases and inter-phrase intervals revealed that this unit's firing was actually suppressed during most twitter phrases, followed by a rebound above spontaneous during early inter-phrase intervals (Fig. 1d). This pattern was distinct from passive sensory responses during playback of an animals' own vocalizations, which showed increases during both twitter phrases and intervals (Fig. 1c, d).

Examining the overall population average activity similarly suggested differences between responses during phrases and inter-phrase intervals, with an average bias towards decreased firing during vocalization (Fig. 1e, f). Focusing on the larger population of vocally-suppressed units (defined by a Response Modulation Index, comparing vocal to pre-vocal firing, of RMI ≤ −0.2, see Methods; $n = 1508$ units) revealed a pattern of strong phasic decreases during individual twitter phrases (Fig. 1e, g). During the inter-phrase intervals, the activity of this population increased slightly, but still exhibited decreases below baseline, suggesting the presence of a global process across the entire vocal duration. In contrast, the smaller population of excitatory units (RMI ≥ 0.1, $n = 393$ units) showed a stronger excitation during vocal phrases than during the intervals (Fig. 1g).

The presence of population average activities exhibiting both sustained changes during vocalization as well as differences between phrases and inter-phrase intervals suggested that there may have been two different time-scales of vocal responses during twitters, one phasic (phrase-specific) and one tonic (global). However, because this was a population-average response it was unclear whether these two components represented distinct neural populations, or whether the two were both present within individual neurons. We therefore sought to quantify these twitter responses by comparing activities during phrases and intervals for individual units (Fig. 2). Comparisons amongst units' phrase and interval responses demonstrated that, while correlated (r = 0.46, $p = 8 \times 10^{-186}$, Pearson correlations with two-sided t-test comparisons) and biased towards suppression (negative RMIs), there was significant variability of responses (Fig. 2a). On average, units suppressed during phrases (RMI ≤ −0.2) also exhibited suppression during intervals, but less so than during the phrases, with an average above the unity line (Fig. 2a, orange). These results could potentially be explained by tonic suppression spanning across the entire twitter call, most evident during the intervals, modified by a phasic suppression during phrases where the activity was further reduced below the tonic level. In contrast, excited units (RMI ≥ 0.1) showed interval activities closer to spontaneous (RMI ~ 0), suggesting their phrase responses were stronger than during their inter-phrase intervals. However, considerable variability between units was evident, including units suppressed during phrases but not during the intervals (Fig. 2b, a-b), units equally suppressed during both (c), units excited during phrases with variable excitation or suppression during intervals (d-e), and even units more suppressed during intervals than during phrases (f). This variability suggests that multiple processes of vocal suppression, or excitation, may be present and combined to varying degrees in different units.

One possible explanation for this inter-unit variability would be a tonic vocal suppression seen during inter-phrases intervals that is then modified by phasic activity during vocal phrases. However, such a simple model may not cleanly separate the different components nor explain the variability seen, as activity during phrases may carry over or influence responses during intervals, including some suppressed unit phrases with rebound of activity above spontaneous during the intervals (i.e. Fig. 1d). Further confounding this separation is the significant population-level correlations between phrase and interval activities (Fig. 2a). Phrase-interval (Phr-Int; 'phasic') differences were similarly correlated with the interval activity (r = −0.49, $p = 2 \times 10^{-100}$). In order to decorrelate these phrase and interval responses, and better represent putative phasic and tonic contributions, we transformed our coordinate axis to compare Phr-Int differences (phasic responses) to

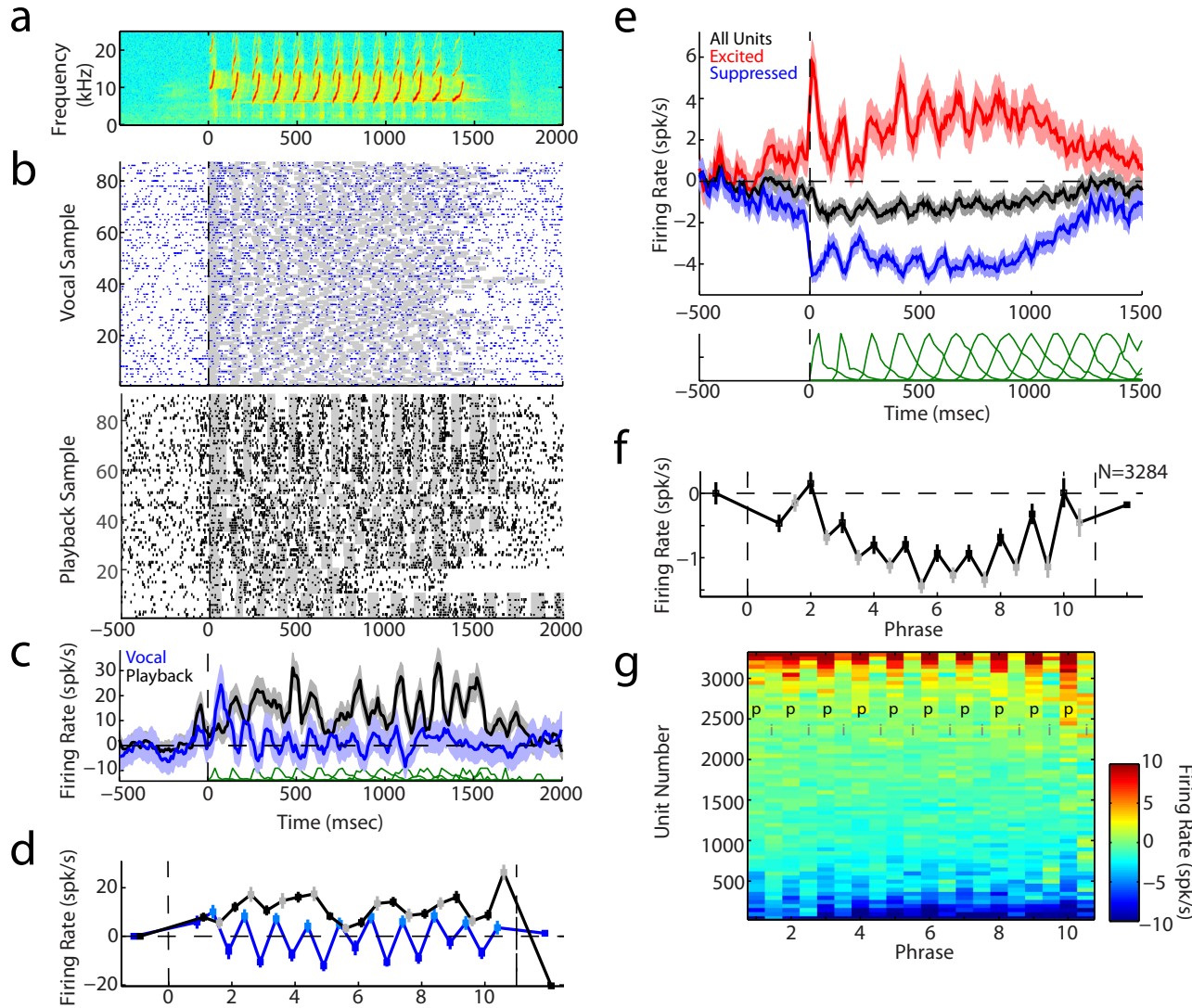

**Fig. 1 | Example and population responses during twitter vocalizations.**
**a** Spectrogram of a marmoset twitter. **b** Raster plot of action potentials before, during, and after twitters recorded from an auditory cortex unit. Responses to vocal production (above) and vocal playback (below) are shown separately. Shaded areas indicate twitter phrases. **c** Vocal-onset aligned peri-stimulus time histogram (PSTH) illustrating mean responses during vocal production (blue, $n = 87$ vocalizations) and playback (black, $n = 90$ vocalizations). Shaded error bars: 95% confidence intervals. Distributions of vocal phrase onset time variability are indicated below (green). **d** Mean firing rate responses during vocal phrases (dark blue) and inter-phrase intervals (light blue) are shown ($n = 87$ vocalizations), as are responses during vocal playback (black/gray, $n = 90$ vocalizations). Error bars: SEM. **e** Population mean PSTHs aligned to vocal onset, shown separately for all units ($n = 3284$ units, black), excited units (Response Modulation Index, RMI $\geq 0.1$;

$n = 393$ units, red), and suppressed units (RMI $\leq -0.2$; $n = 1508$ units; blue). Shaded: bootstrapped 95% confidence intervals. Below: distributions of vocal phrase onsets. **f** Population mean responses showing global suppression with dynamics between phrases (black) and intervals (gray). SEM (error bars) and significant responses (filled symbols indicate points with $p < 0.05$, two-tailed Wilcoxon signed-rank test with FDR correction, exact p-values in Source Data file) are indicated. **g** Firing rate responses during phrases and intervals are shown for the entire neural population, sorted from most suppressed to most excited. Suppressed units (bottom) showed lower firing rates during phrases than intervals, while excited units (top) showed the opposite pattern. For better visualization, units have been averaged into blocks of 50. Color scale has been limited to the [−10 10] range, phrases (p) and intervals (i) are indicated. Source data are provided as a Source Data file.

the phrase-interval averages used as an alternative measure of tonic responses (Fig. 2c). This analysis showed more balanced phasic responses, but with clear biases towards negative differences in suppressed units and positive differences in excited ones (Fig. 2c, right). The diversity of these responses suggests that the two components were present within individual units, but combined to different degrees for different units, including mixing of positive (excitatory) or negative (suppressed) phasic responses with tonic responses of the opposite sign.

The presence of a tonic response spanning across multiple phrases suggests the possibility of a motor plan that similarly encompasses

the entire vocal duration. While such global motor plans have been suggested for other multi-phrase primate vocalizations[31], evidence for twitter calls is lacking. We therefore examined the acoustic structure of marmoset twitters, focusing on the mean frequency, loudness (sound pressure level, SPL) and duration of phrases. Consistent with previous reports[32], we found a regular progression of frequency across twitter phrases (Supplementary Fig. 1c). We further measured correlations between twitter phrases (normalized to exclude differences between animals), and found significant correlation of acoustics between phrases, with later phrases being predictable based upon earlier phrases (Supplementary Fig. 1d, e). Notably, we found that the mean

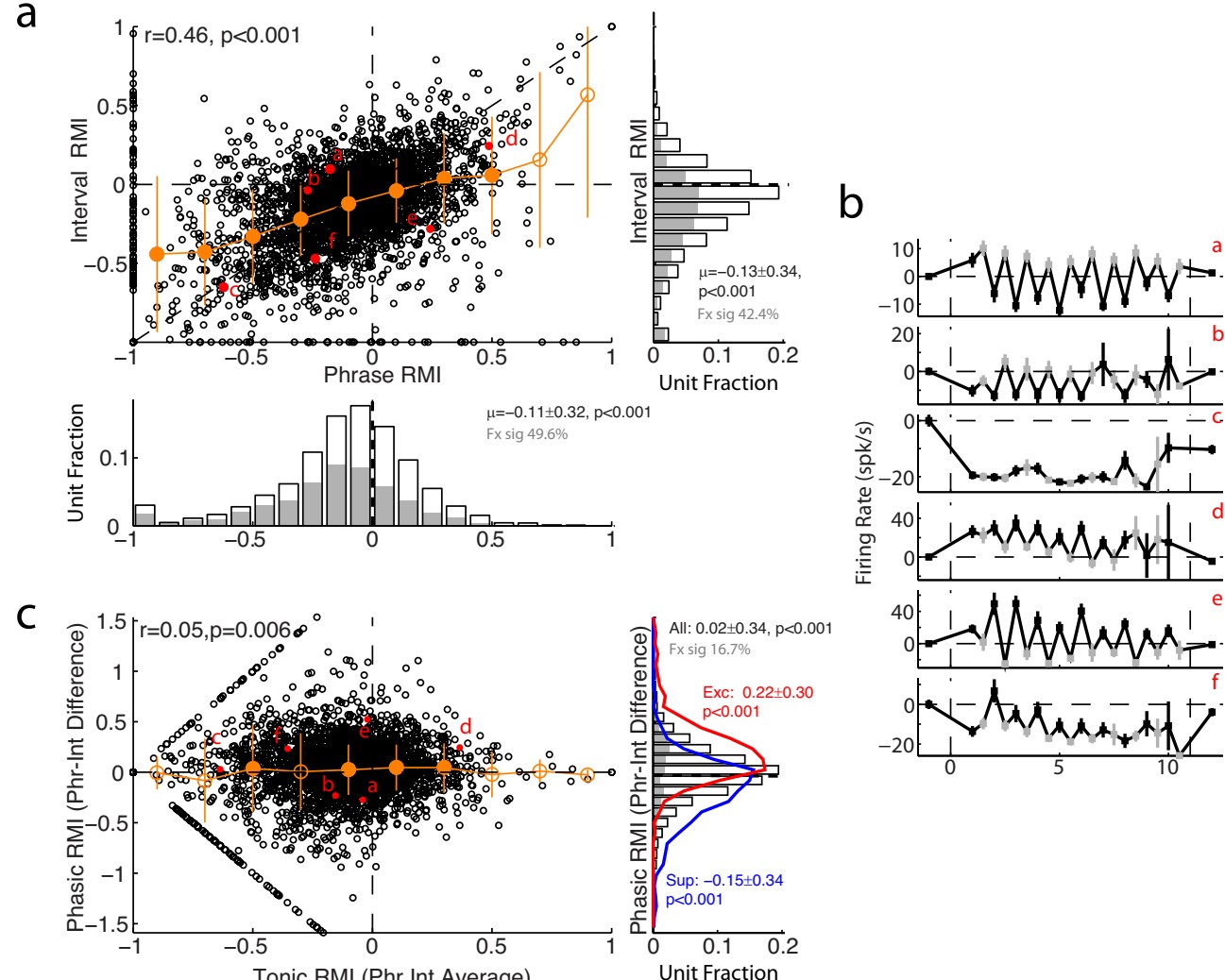

**Fig. 2 | Separability of phasic and tonic vocal responses. a** Scatter plot comparing phrase and interval RMIs for all units. Correlation coefficient and p-value indicated ($p = 8 \times 10^{-186}$, Pearson correlation and two-sided t-test, $n = 3284$ independent units). Interval responses, binned by phrase RMIs (orange: mean ± SEM, filled symbols: $p < 0.05$, two-sided signed-rank tests with FDR corrections, exact p-values in Source Data file), show a bias towards less interval suppression or excitation for suppressed and excited neurons (closer to 0), respectively. Distributions for phrase (bottom) and interval (right) RMIs are shown, along with their respective mean±s.d., demonstrating significant bias towards negative responses ($p = 6 \times 10^{-141}$ and $6 \times 10^{-86}$, two-sided signed-rank test, $n = 3284$ units). Units individually significant are indicated (filled bars, $p < 0.05$, two-sided signed-rank tests) as is their fraction of the total population ('Fx sig'). Labeled units are examples shown in (**b**). **b** Representative sample units illustrating the diversity of phrase and interval responses. Phrase (black) and interval (gray) means are shown as SEMs ($n = 87, 16, 33, 11, 19$, and $37$ vocalizations, respectively). Example '*a*' is the same unit from Fig. 1a. **c** Scatter plot comparing phasic (Phr-Int differences) against tonic (average of phrases and intervals) vocal responses, showing a decorrelation from the Phr-Int difference ($p = 0.006$; Pearson correlation and two-sided t-test, $n = 2384$ units). Phasic differences are binned by tonic RMI, as in **a** (orange, mean ± SEM). Distribution for phrase-interval difference is also shown (right), overlayed with distributions sorted into suppressed (RMI ≤ −0.2, blue) and excited (RMI ≥ 0.1, red) units, showing bias towards negative differences in suppressed and positive in excited populations (all: $p = 4 \times 10^{-11}$, $n = 2384$ units, suppressed: $p = 1 \times 10^{-41}$, $n = 1008$, excited: $p = 2 \times 10^{-72}$, $n = 686$; two-sided signed-rank tests). Source data are provided as a Source Data file.

frequency of the first third of twitter phrases significantly predicted the middle ($r = 0.60$, $p < 1 \times 10^{-200}$, Pearson correlations with two-sided t-tests, $n = 13,804$ twitters) and last third ($r = 0.46$, $p < 0.001$) of phrases, with similar relationship between middle and final thirds ($r = 0.61$, $p < 1 \times 10^{-200}$). Similar correlations were noted for phrase SPL ($r = 0.87$, $0.78$, and $0.90$, respectively, $p < 1 \times 10^{-200}$) and phrase duration ($r = 0.71$, $0.54$, and $0.67$, $p < 1 \times 10^{-200}$). These positive correlations demonstrate that when earlier phrase acoustics were above or below average (for a given animal and phrase), subsequent phrases were also above/below average. This predictability of later phrases, based upon earlier phrases, is similar in magnitude to past studies, which have been interpreted as evidence of a larger motor plan during multi-phrase vocal production[31]. We also found that the

number of twitter phrases produced was weakly predictable based upon the frequency and SPL of early phrases (frequency: $r = 0.15$, $p = 6 \times 10^{-71}$; SPL $r = 0.21$, $p = 2 \times 10^{-143}$, Pearson correlations and two-sided t-test).

In order to exclude the possibility that the apparent separability of tonic and phasic responses was due to the mixing of different individual units within multi-units, we repeated our analysis of phasic and tonic responses for single-units and found similar results (Supplementary Fig. 2). We also performed a principal component analysis of phrase and interval responses across the population and found a similar decomposition between tonic (first component) and phasic (second and third components) responses, suggesting their separability (Supplementary Fig. 3). Interestingly, these principal

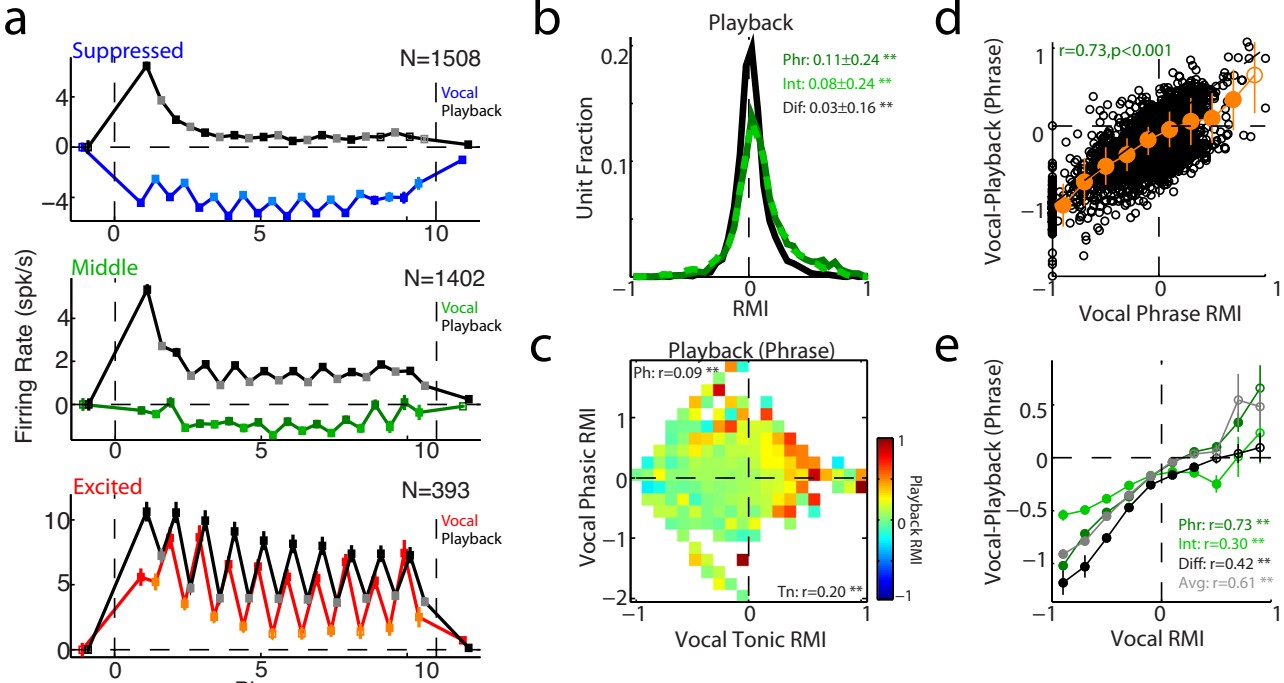

**Fig. 3 | Comparison of vocal and auditory responses. a** Population mean responses for suppressed (RMI ≤ −0.2, top), middle (−0.2 < RMI < 0.1, middle), and excited (RMI ≥ 0.1, bottom) units comparing vocal and playback responses. Phrases (dark blue, dark green, red) and inter-phrase intervals (light blue, light green, orange) are indicated. Average responses to passive vocal playback are shown for their respective unit populations (phrase: black, interval: gray). Error bars indicate SEM. Significant bins are indicated by filled symbols ($p < 0.05$ two-tailed Wilcoxon signed-rank test with FDR correction, exact p-values in Source Data file). **b** Distribution of responses during playback, sorted by phrase (Phr, dark green), interval (Int, light green), and phasic Phr-Int differences (Dif, black) showing a positive response, in contrast to suppressed vocal production responses. Mean and standard dev are indicated as are significant responses (** $p < 0.001$; Phr: $p = 2 \times 10^{-4}$, Int: $p = 2 \times 10^{-89}$, Dif: $p = 2 \times 10^{-29}$, two-sided signed-rank tests, $n = 3243$ units). **c** Mean playback responses are compared to vocal phasic and tonic activity. Units have been binned by tonic and phasic RMI (as in Fig. 2c), and then the playback RMI (measured during playback phrases) is averaged for those units. The color indicates the mean playback RMI for a given bin. Partial correlation

coefficients are indicated, as is the significance (** $p < 0.001$; Ph: phasic, $p = 6 \times 10^{-7}$; Tn: tonic, $p = 4 \times 10^{-31}$, two-sided t-test, $n = 3243$ units). **d** Scatter plot comparing vocal-playback differences against vocal phrase responses (r = 0.73, $p < 1 \times 10^{-200}$, Pearson correlation coefficient and two-sided t-test, $n = 3243$ units). Suppressed neurons (negative vocal RMIs) exhibited decreased activity compared to playback, while excited units (positive RMIs) had more equivalent or increased vocal responses (orange: mean ± SEM, filled symbols: $p < 0.05$, two-sided signed-rank test with FDR correction). **e** Mean vocal-playback differences measured during phrases as in **d** (mean ± SEM, $n = 3243$ units), but binned separately based on how the vocal response RMI was measured: during phrases (dark green), intervals (light green), phasic Phr-Int difference (black), or tonic average (gray). Vocal-playback differences were greater for phrases and less for the intervals (closer to zero difference). Phasic responses showed the largest decrease for negative RMI units (suppressed), but had little difference between production and playback for positive RMI units (excited). Asterisks indicate significant correlations (Phr: $<1 \times 10^{-200}$, Int: $3 \times 10^{-70}$, Diff: $1 \times 10^{-140}$, Avg: $<1 \times 10^{-200}$; Pearson correlations with two-side t-tests, $n = 3243$ units). Source data are provided as a Source Data file.

components also showed a temporal evolution over successive phrases/intervals of unclear significance.

## Sensory tuning and tonic/phasic responses

The presence of two separate vocal responses, one phasic and one tonic, raises important questions as to the respective origins of these two components. We examined to what extent passive auditory responses and frequency tuning could offer an explanation. Consistent with previous results[10,33], we found that suppressed units exhibited decreased vocal production activity compared to passive playback of vocal sounds, while excited neurons had more similar responses (Fig. 3a). However, unlike vocal production, where responses were largely suppressed (i.e. Fig. 2a), responses to twitter playback were biased towards excitation both during phrases and intervals, with positive phrase-interval differences (Fig. 3b). Unit by unit comparisons of vocal production and playback responses showed the strongest playback activity amongst vocally excited units, both for those with excited phasic responses (positive Phr-Int difference, Fig. 3c y-axis) and those with excited average/tonic responses (Fig. 3c x-axis), findings consistent with previous suggestions of auditory cortical

excitation during vocalization as a passive sensory response[10,33]. Overall, responses during vocal production were decreased compared to playback, particularly for vocal suppressed neurons (negative RMI) and less so for vocal excitation (Fig. 3d, e). Importantly, this relationship was weaker for inter-phrase intervals, where vocal-playback differences were closer to zero, consistent with the presence of auditory inputs evoking excitatory responses during twitter phrases, but not during silent intervals. Phasic responses also showed this pattern, with large vocal-playback decreases in suppressed units, but no vocal-playback difference in excited ones (Fig. 3e). Interestingly, even units with little modulation during vocal production, vocal RMI ~ 0, showed decreases between vocal production and playback, suggesting they are also subjected to a degree of vocal suppression (Fig. 3a middle, Fig. 3e). These negative differences between vocal production and playback were seen both during vocal phrases (Fig. 3e) as well as when measured for differences during the intervals and tonic averages (Supplementary Fig. 4). Interestingly, the phasic component was often stronger during vocal production than playback for those excited units (positive vocal RMI, Supplementary Fig. 4g). Such observations suggest sensory responses alone cannot explain either tonic or phasic

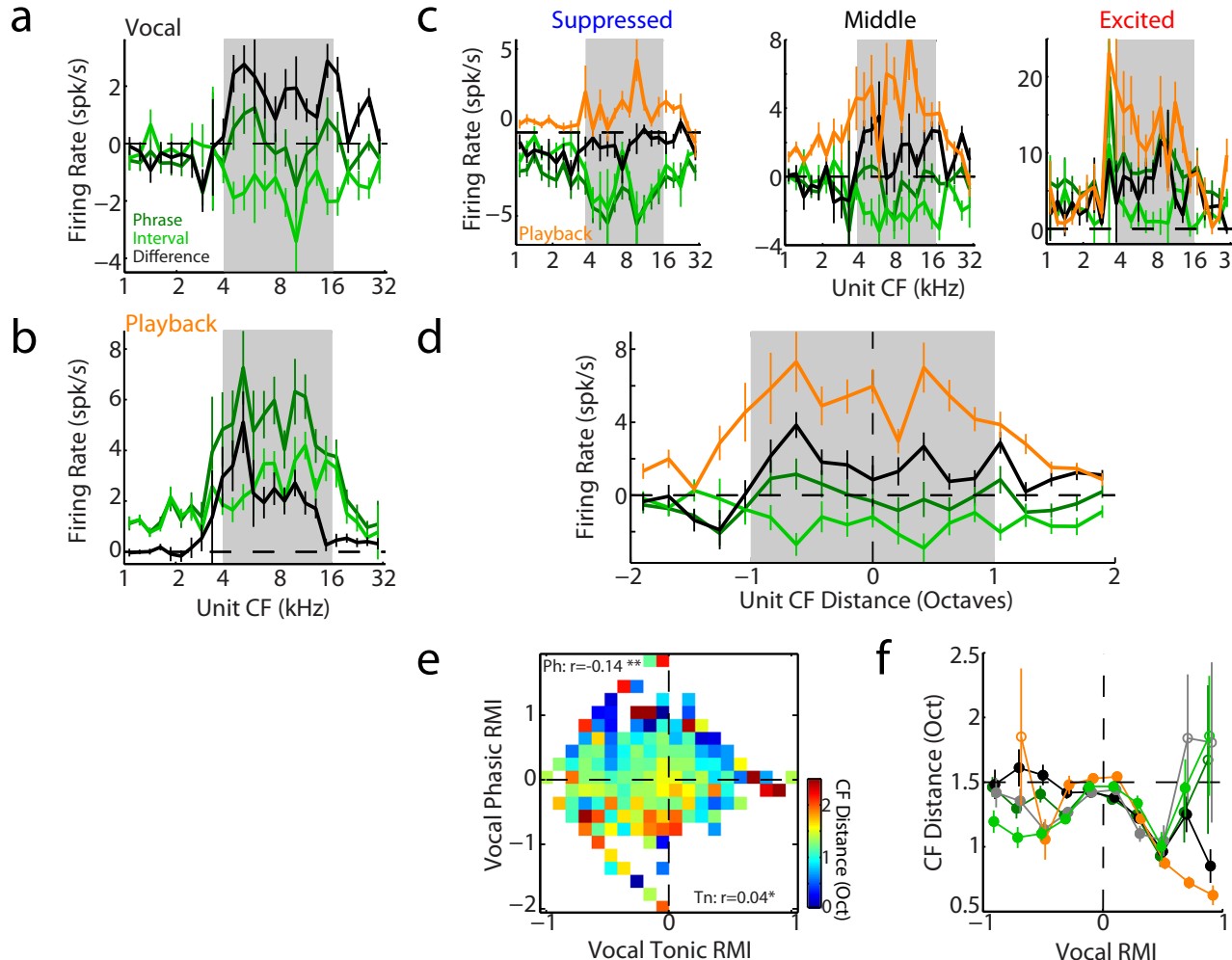

**Fig. 4 | Comparison of vocal responses and frequency tuning. a, b** Average vocal (**a**) and playback (**b**) responses binned by units' center frequency (CF) tuning. Mean ± SEM are shown ($n = 2136$ units each). Both vocal interval (light green) suppression and phrase (dark green) excitation exhibited frequency dependence that matched the range of twitters' fundamental frequency (shaded), as did the phrase-interval differences (black). **c** CF dependence sorted by suppressed (RMI ≤ −0.2; left, $n = 870$ units), middle (−0.2 < RMI < 0.1; middle, $n = 952$ units), and excited (RMI ≥ 0.1; right, $n = 314$ units) units, showing frequency-tuned interval suppression and phrase/phasic excitation. Middle units exhibited mixed phrase responses with excitatory phasic Phr-Int differences combined with interval suppression. Suppressed units showed decreases during phrases, intervals, and Phr-Int differences; the latter were less frequency specific. Coloring as in **a**. Playback responses for respective populations are also shown (orange). **d** Average vocal responses are shown compared to a unit's frequency distance relative to a vocal reference frequency (7.5 kHz). Units with more distant CFs exhibited less vocal or playback responses, particularly out of the shaded ±1 octave range. Mean ± SEM are shown ($n = 2136$ units, coloring as in **a**). **e** Average unit CF distance (absolute value) is shown relative to binned tonic and phasic vocal responses, as in Fig. 3c, showing narrower CF distances for some units with phasic excitation. Partial correlation coefficients are indicated (*$p < 0.05$, **$p < 0.001$; Ph: $p = 5 \times 10^{-10}$, Tn: $p = 0.040$, two-sided t-tests, $n = 2136$ units). **f** Average unit CF distances are shown binned by vocal RMI separately for phrase, interval, tonic (gray), and phasic vocal responses, as well as playback responses. Playback and phasic responses showed smaller distances from vocal frequencies for excited (positive RMI) compared to suppressed (negative RMI) units. Interval and tonic responses exhibited more non-linear relationships to RMI, with narrower tuning ranges for suppressed units that increased for middle units, and then decreased again for excitation. Mean ± SEM are shown, filled symbols: $p < 0.05$ (two-sided signed-rank tests with FDR correction, $n = 2136$ units, exact p-values in Source Data file). Source data are provided as a Source Data file.

vocal suppression, though phasic vocal excitation correlates with that seen during playback and may be explained by excitatory sensory inputs.

We also examined to what extent tonic and phasic vocal responses could be related to frequency tuning (center frequency, CF) of these auditory units. Population average responses showed stronger suppression during inter-phrase intervals for units overlapping the frequency range of twitter fundamental frequencies (4–16 kHz; Fig. 4a) than for lower or higher CF units. Phrase responses and Phr-Int differences were similarly frequency tuned, but more excitatory, suggesting a population-average phasic (auditory) excitation combined with tonic suppression. Playback responses also exhibited similar excitatory frequency dependence, particularly for phrase-interval

differences (Fig. 4b). Breaking apart the neural population into suppressed, middle, and excited units (based on vocal phrase responses) confirmed the overall pattern of suppression during vocal intervals (Fig. 4c). Interestingly, the middle units, which would have previously been classified as unresponsive (RMI ~ 0), best demonstrated a combination of tonic suppression, seen during the intervals, and phasic excitation during phrases canceling each other (Fig. 4c, middle). These same units showed excitation during playback (Fig. 3a), but would not have had the superimposed suppression during playback that was only seen during vocal production. This suggests that many of these 'unresponsive' units do actually exhibit an element of vocal suppression. In contrast, suppressed units showed frequency-dependent decreases during both phrases and intervals (Fig. 4c, left). This

frequency specificity was more notable during the intervals than during phrases or Phr-Int differences which showed a more broadly tuned decrease.

In order to better understand the frequency specificity of these different tonic and phasic components, we calculated the frequency distance of unit CFs from a reference frequency in the middle of the vocal range (7.5 kHz; Fig. 4d). We found that both playback responses and phasic differences decreased with CF distance from the reference frequency ($r = -0.28$, $p = 1 \times 10^{-89}$ and $-0.13$, $p = 9 \times 10^{10}$, Pearson correlations with two-sided t-test corrections, $n = 2136$ units), while interval responses increased ($r = 0.1$, $p = 2 \times 10^{-6}$) and phrase responses were flat ($r = -0.04$, $p = 0.09$) across this range. Multi-variate linear regression showed a significant frequency dependence for the playback ($-0.08$, 95% confidence intervals $[-0.10 \; -0.06]$, $p = 5 \times 10^{-22}$), phasic differences ($-0.05$ $[-0.06 \; -0.03]$, $p = 5 \times 10^{-8}$) and intervals ($0.03$ $[0.02 \; 0.05]$, $p = 8 \times 10^{-5}$). We further examined if this CF dependence might be influenced by frequency tuning bandwidth, and found larger bandwidths for tonically excited units, and stronger vocal excitation for units with higher bandwidths near vocal frequencies (Supplementary Fig. 5). Normalizing the CF distance metric by a unit's bandwidth slightly tightened the frequency dependence of auditory and phasic excitation, as well as interval suppression (Supplementary Fig. 5c).

Because both phasic and tonic vocal responses varied widely between units, including both positive and negative values, we also sought to disambiguate how frequency dependence might be different for suppression vs. excitation. Comparing the CF distance from the reference vocal frequency (absolute distance) to tonic and phasic responses showed significant correlations for both, but stronger for the phasic ($r = -0.14$, $p = 5 \times 10^{-10}$, Pearson and two-sided t-test, $n = 2136$) than tonic ($r = 0.04$, $p = 0.046$), an inverse correlation suggesting closer frequency tuning in units with phasic excitation than phasic suppression (Fig. 4e).

Further comparisons between vocal responses and CF distance showed that these dependances were often non-linear (Fig. 4f), but did show significant differences when comparing CFs of units with suppressed, middle, and excited RMIs for phasic, tonic, interval and auditory responses (multivariate ANOVA, df=2133; phasic: F = 4.66, $p = 0.01$; tonic: F = 3.52, $p = 0.03$; interval: F = 6.76, $p = 0.001$; playback: F = 81.54, $p = 7 \times 10^{-35}$). Notably, while CFs were closer to vocal frequencies (lower CF distances) for units with excitatory phasic and playback responses, as expected for sensory responses, interval suppression was more frequency-specific than phasic suppression (light green vs. black at negative vocal RMIs in Fig. 4f; ANOVA F = 2.71, df=2133, $p = 0.043$). Overall, these results suggest that both phasic and tonic/interval vocal suppression are frequency-specific, but more so for the tonic/interval component. The presence of a mix of tonic suppression and phasic excitation, both frequency dependent, in many units may also explain the apparent lack of frequency-tuned vocal suppression in our previous studies[33]. Phrase responses, which most closely matches our previous analyses of longer vocalizations, combine both tonic and phasic responses, did not have CF distances that significantly vary between units with suppressed, middle, or excited phrases (Fig. 4f, multivariate ANOVA F = 2.19, df=2133, $p = 0.11$). In contrast, intervals, phasic, tonic, and auditory responses clearly did vary with CF.

Because vocal frequencies are also not static across a vocalization or vocal phrase, we performed one additional analysis to compare the effects of frequency tuning on vocal and playback responses (Supplementary Fig. 6). A spike phase analysis was used to measure the relative timing of each spike during twitter phrases and intervals, using each phrase-interval pair as a single circular cycle. For suppressed units, where spiking occurred more often during the inter-phrase interval, spikes were distributed with higher phase values than for excited units where responses were more synchronized at earlier

phases during vocal phrases (Supplementary Fig. 6a). Population distributions of unit mean phase were skewed towards later phases during vocal production than during playback, indicating spikes occurring mainly during the intervals for production and more during the phrases for playback (Supplementary Fig. 6b). This difference was likely due to the large number of vocally suppressed neurons, which have less spiking during phrases, as differences in population averages between vocal suppression and excitation were also seen (Supplementary Fig. 6c). Comparing mean phase during playback to unit frequency tuning showed a progression over time, with earlier responses for lower CF units in the vocal range, similar to the spectro-temporal dynamics of twitter phrases (Supplementary Fig. 6d, $r = 0.10$, $p = 0.003$, Pearson correlation and two-side t-test). Similar timing was not seen during vocal production ($r = -0.02$, $p = 0.68$), where mean phases were later during the inter-phrase intervals. Dividing positive and negatively phasic vocal responses showed earlier response times in phasic excitation compared to suppression ($2.97 \pm 2.19$ radians vs. $4.86 \pm 0.60$, $p = 8 \times 10^{-25}$, two-sided rank-sum test, $n = 471$ and 382 units, respectively), a difference that was not seen when comparing positive and negative tonic responses ($4.2 \pm 1.8$ vs $4.0 \pm 1.73$, $p = 0.03$), and suggestion of temporal dynamics similar to vocal playback. These results are consistent with the idea of phasic, more than tonic, excitation as a sensory response.

### Pre-vocal suppression, onset responses, and other call-types

The question remains as to possible origins of tonic and phasic vocal suppression. One of the strongest arguments for vocal suppression as a motor phenomenon, rather than a purely sensory one, has been the presence of suppression prior to the onset of vocalization[10,34]. A similar pre-vocal onset of suppression was seen for twitters (Fig. 1e). We therefore compared pre-vocal responses to those during vocalization and found a population-level correlation with tonic responses (Fig. 5a, $r = 0.22$, $p = 4 \times 10^{-37}$, Pearson correlations and two-sided t-test, $n = 3280$ units). In contrast, this correlation with pre-vocal suppression was absent for phasic Phr-Int differences ($r = 0.01$, $p = 0.65$). Trial-to-trial correlations of pre-vocal and vocal activity, calculated individually for each unit, exhibited a similar pattern with 19.7% of units showing significant correlations (Fig. 5b). Significantly stronger pre-vocal correlations were seen for tonic average responses than either phrases or intervals alone (Kruskal Wallis ANOVA, $\chi^2 = 2.3 \times 10^7$, $n = 2489$ units, $p = 0.002$). These unit pre-vocal correlations were also stronger in suppressed than excited units ($0.22 \pm 0.36$ vs. $0.18 \pm 0.30$, $p = 0.006$, two-sided rank-sum test, $n = 673/490$ units), again correlating with tonic suppression (Supplementary Fig. 7). This correlation with pre-vocal responses suggests that tonic suppression begins in anticipation of vocal production, and is then subsequently modified by phasic activity, either excitatory or inhibitory, during vocal phrases.

Because auditory responses often exhibit onset activity followed by adaptation or synaptic depression, we also examined the relationship between firing rates of the first twitter phrase and subsequent phrases or intervals. We found significant correlations between the first phrase and subsequent phrases, intervals, as well as tonic and phasic responses (Fig. 5c). This was present both for responses during vocal production as well as playback. Notably, we saw very few units that exhibit strong excitatory first phrase activity followed by decreased or suppressed activity during the remainder of the call. These results suggest that observed twitter responses were likely not a simple result of onset excitation followed by adaptation, which might be expected to have an inverse correlation.

In order to further reconcile tonic and phasic responses components to our previous results, we also compared our twitter responses to those during other marmoset call types we previously studied (phee, trill, trillphee)[34] and found similar responses within individual units and bias towards suppression (Fig. 5c). Responses during these

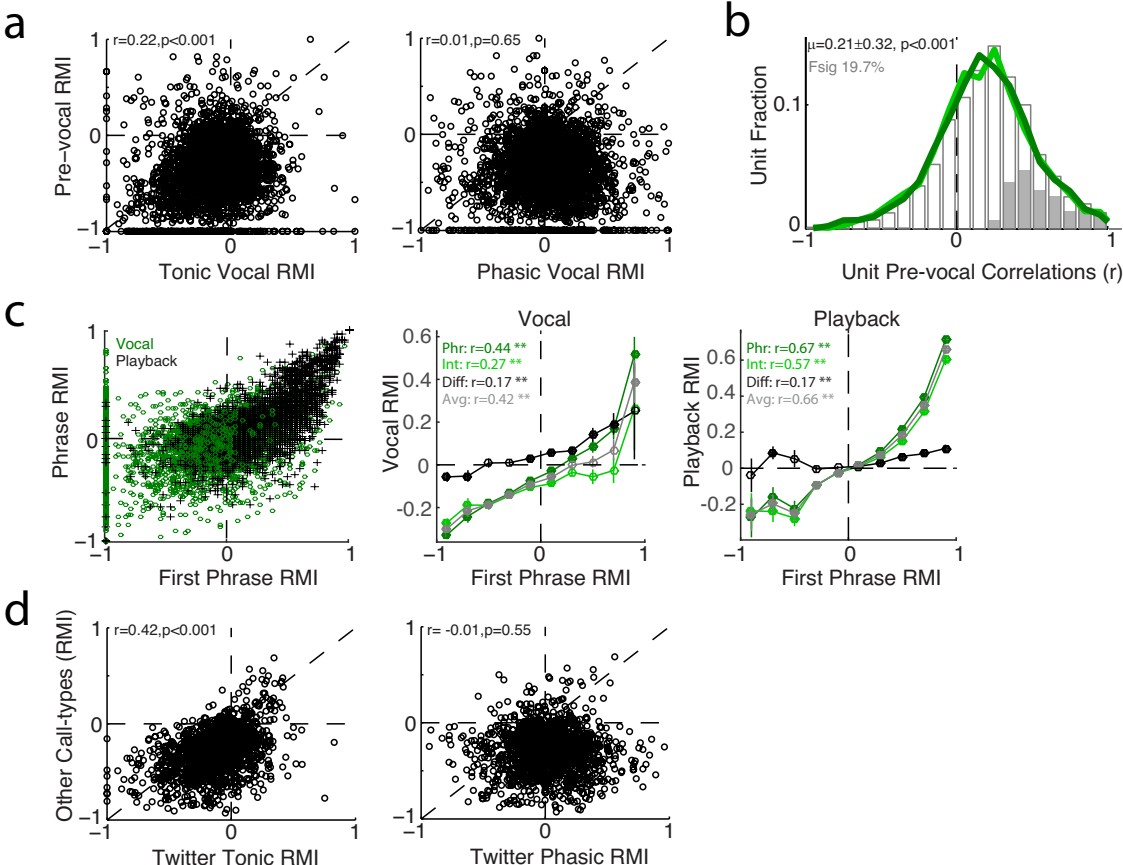

**Fig. 5 | Comparison to pre-vocal suppression, onset responses, and other types of marmoset calls. a** Scatter plot showing a significant correlation between average/tonic vocal responses (left) and activity in the pre-vocal period (−200 to 0 ms), but not for pre-vocal activity and phasic responses (right). Pearson correlations are indicated (Tonic: $p = 4 \times 10^{-37}$, Phasic: $p = 0.65$, two-sided t-tests, $n = 3280$ units). **b** Distribution of correlation coefficients between tonic vocal and pre-vocal activity, calculated across trials for individual units, showing a significant bias towards positive correlations (mean±s.d. indicated; $p = 7 \times 10^{-179}$, two-sided signed-rank test, $n = 2489$ units). Fraction of individually significant neurons indicated (shaded, $p < 0.05$, Pearson correlations and two-sided t-tests). Similar distributions were noted for pre-vocal correlations with phrases (dark green) and intervals (light green), though the correlations with tonic average responses were stronger ($\chi^2 = 2.8 \times 10^7$, $n = 2489$ units, $p = 0.023$, Kruskal Wallis ANOVA). **c** Scatter plot comparing responses during the first twitter phrases and the average of subsequent phrases for vocal production (green; r = 0.44, $p = 3 \times 10^{-154}$, Person correlation with

two-sided t-test, $n = 3280$ units) and playback (black r = 0.67, $p < 1 \times 10^{-200}$). Mean vocal responses during phrases and intervals, binned by onset response, are shown for vocal production (middle) and playback (right). Mean ± SEM are shown ($n = 3280$ units). There were significant correlations between the response to the first phrase and responses during subsequent phrases and intervals (** $p < 0.001$; Vocal: Phr $p = 3 \times 10^{-154}$, Int $p = 3 \times 10^{-56}$, Diff $p = 1 \times 10^{-23}$, Avg $p = 2 \times 10^{-139}$; Playback: $p < 1 \times 10^{-200}$, $p < 1 \times 10^{-200}$, $p = 1 \times 10^{-23}$, and $p < 1 \times 10^{-200}$, respectively; Pearson correlations with two-sided t-tests. **d** Scatter plot comparing responses for each unit between twitters and other marmoset call-types. Significant correlations were seen for tonic (left), but not phasic (right) twitters responses ($p = 8 \times 10^{-74}$ and $p = 0.55$, respectively; Pearson correlations with two-sided t-tests, $n = 1663$ units). Though correlated, responses during twitters were significantly less suppressed than during the other call types (RMI difference +0.16 ± 0.24, $p = 3 \times 10^{-128}$ two-sided signed-rank, $n = 1663$). Source data are provided as a Source Data file.

longer vocalizations were more strongly correlated with tonic than with phasic twitter suppression.

## Anatomic variation of tonic and phasic responses

The presence of separable tonic and phasic vocal responses may also explain previous inconsistencies in anatomic correlates of vocal suppression. Previous work in marmosets has suggested a slight increase in vocal feedback sensitivity in the right hemisphere, as well as greater vocal behavioral effects of microstimulation, but no differences in net vocal suppression[16]. Comparing average/tonic twitter responses between hemispheres showed a small bias towards greater suppression on the right (Fig. 6a), with greater right-sided suppression during intervals, but not Phr-Int differences (Fig. 6b). This suggests a hemispheric bias for tonic suppression, but not for phasic responses, and may explain why phrase responses, which combined tonic and phasic components, were not different between hemispheres.

Similar comparisons based upon the electrode row in our multi-electrode arrays showed greater average suppression in more lateral

electrodes (Fig. 6c). The largest effect was for phasic Phr-Int differences, where lateral electrode responses were decreased compared to those more medially. This difference appears best explained by stronger phasic excitation in the most medial electrode row (presumably primary auditory cortex, A1), with more balanced phasic suppression and excitation in more lateral rows (lateral A1 or auditory belt; Fig. 6d). Further comparisons of playback responses between electrode rows showed stronger activity in more medial rows, particularly for phasic excitation, further supporting stronger phasic excitation in A1 (Fig. 6e). Tonic responses also varied between rows, but to lesser degree than phasic. Overall, tonic vocal suppression appears stronger in areas likely to be higher-order auditory cortex, while phasic excitation appears stronger in primary cortex.

## Feedback-sensitivity during tonic and phasic vocal suppression

Finally, to understand if these two components of vocal suppression could have different functional roles in self-monitoring behaviors, we examined responses during frequency-shifted vocal feedback.

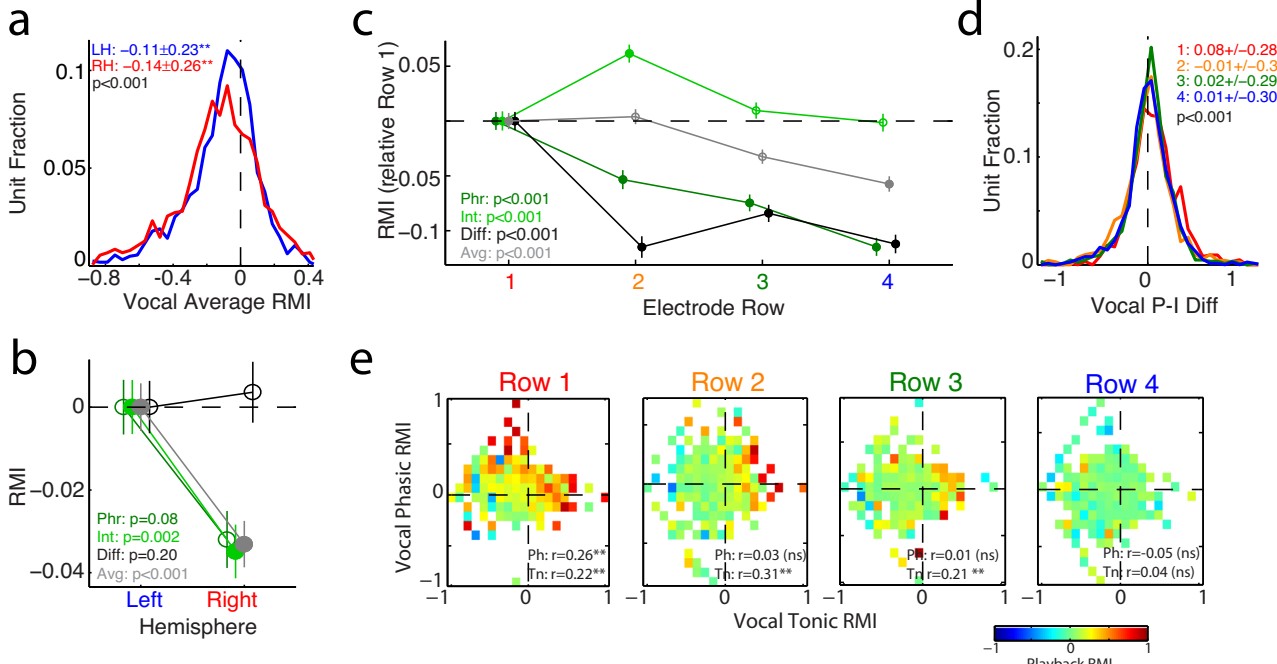

**Fig. 6 | Anatomical variation in twitter responses. a** Histograms comparing average vocal response between right (RH, red) and left hemispheres (LH, blue). Both hemispheres showed significant average suppression (** $p < 0.001$; LH: $n = 1634$ units, $p = 3 \times 10^{-73}$, RH: $n = 1650$, $p = 2 \times 10^{-85}$; two-sided signed-rank test), but stronger on the right ($p = 3 \times 10^{-4}$, two-sided rank-sum test). **b** Comparison of phrase, interval, phasic Phr-Int difference, and average/tonic activity between hemispheres. Mean and SEM are shown, results have been normalized to the left hemisphere mean response (unit numbers as in **A**). Significantly stronger right hemisphere suppression was noted for tonic responses (filled symbols), reflected in the interval and average activity ($p = 0.002$ and $p = 3 \times 10^{-4}$, respectively, two-sided rank-sum tests), but not the phrases or Phr-Int differences. **c** Comparison of responses across electrode rows. To account for variably placement, responses have been normalized to the mean response for the most medial row in each electrode array, likely corresponding to A1. Mean ± SEM shown ($n = 663, 750, 935$, and 807 units for rows 1–4, respectively). Significant trends towards greater

suppression in more lateral electrodes (likely lateral A1 or lateral belt/parabelt) were noted, except interval activity, but all showed significant variations between rows (Phr: $p = 4 \times 10^{-11}$, Int: $p = 5 \times 10^{-6}$, Diff: $p = 5 \times 10^{-13}$, Avg: $p = 1 \times 10^{-7}$; Kruskal-Wallis ANOVAs). Phr-Int differences showed the greatest changes between the first row and more lateral electrodes, the distributions of which are shown in (**d**) (Mean and std are indicated for rows 1–4, $\chi^2 = 34.3$, $p = 2 \times 10^{-7}$, Kruskal Wallis). **e** Comparison of playback responses across electrode rows, sorted by vocal tonic/phasic activity. Playback responses showed phasic (Phr-Int difference) vocal correlations only in row 1 and tonic (average) correlations in rows 1–3, consistent with stronger vocal phasic excitation in row 1 being due to sensory inputs. Significant correlations are indicated (** $p < 0.001$; ns: non-significant; Row1: $n = 642$, Ph/Tn: $p = 1.6 \times 10^{-11}/2 \times 10^{-8}$; Row2: $n = 746$, $p = 0.39/7 \times 10^{-18}$; Row3: $n = 926$, $p = 0.70/8 \times 10^{-11}$; Row 4: $n = 801$, $p = 0.16/0.30$; partial correlations with two-sided t-tests). Source data are provided as a Source Data file.

Previous work has shown auditory cortex neurons to be sensitive to shifted feedback during non-twitter vocalizations[13,16]. We found that shifted feedback reduced vocal suppression during both phrases and inter-phrase intervals for suppressed units (Fig. 7a). Shifted feedback did not have as strong an effect on phrase activity for excited units, consistent with previous work, but did increase firing during their inter-phrase intervals, though not significantly. This feedback sensitivity was seen during both phrases and intervals and correlated with tonic suppression (i.e. positive correlation coefficient with tonic RMI, Fig. 7b, c). Interestingly, feedback had an inverse relationship with phasic responses during phrases and intervals. Units with phasic suppression (negative phasic RMI) exhibited increased feedback activity during vocal phrases, but primarily for those with co-existing tonic suppression (having both negative phasic and tonic RMIs). In contrast, those units with phasic excitation showed feedback increases during the intervals (regardless of the tonic response). This flipped dependence of feedback effects upon phasic responses can best be seen when comparing feedback responses between phrases and intervals across different RMIs (Fig. 7c). During phrases, all feedback differences have a flat or negatively sloping relationship to RMI, however during the intervals, only the phasic response (black) changes to a positive slope. The cause of this differential feedback effect based on phasic response is unclear. One possibility is that because phasic excitatory responses are likely sensory, this interval activity could reflect hardware delays placing feedback acoustic

energy in the inter-phrase interval. Alternatively, these feedback patterns could be the result of bias due to a floor effect (i.e. zero activity during normal phrases/intervals that could, therefore, only increase during shifted feedback). We compared feedback responses between low spontaneous ($\leq 5$ spk/sec) and high spontaneous units, and found qualitatively similar patterns of feedback sensitivity regardless of spontaneous rate, suggesting this could not account for feedback effects (Supplementary Fig. 8). Overall, these results suggest a more consistent relationship between feedback sensitivity and tonic rather than phasic vocal suppression, but with stronger effects when both suppressions were combined.

## Discussion

In this study, we examined the possibility of multiple contributions to vocalization-induced suppression in auditory cortex by exploiting the phrasic structure of marmoset twitter calls. We found that there were two distinct components of vocal suppression, operating on different time scales and with potentially different neural mechanisms and functions. The first was a phasic suppression only present during individual vocal phrases; the second was a tonic change beginning before vocal onset and maintained throughout the duration of call. Individual neurons appeared to combine these two suppressive components with auditory feedback to varying degrees, resulting in considerable diversity of observed vocal sensory-motor responses and suggesting a greater degree of complexity to vocal motor predictions

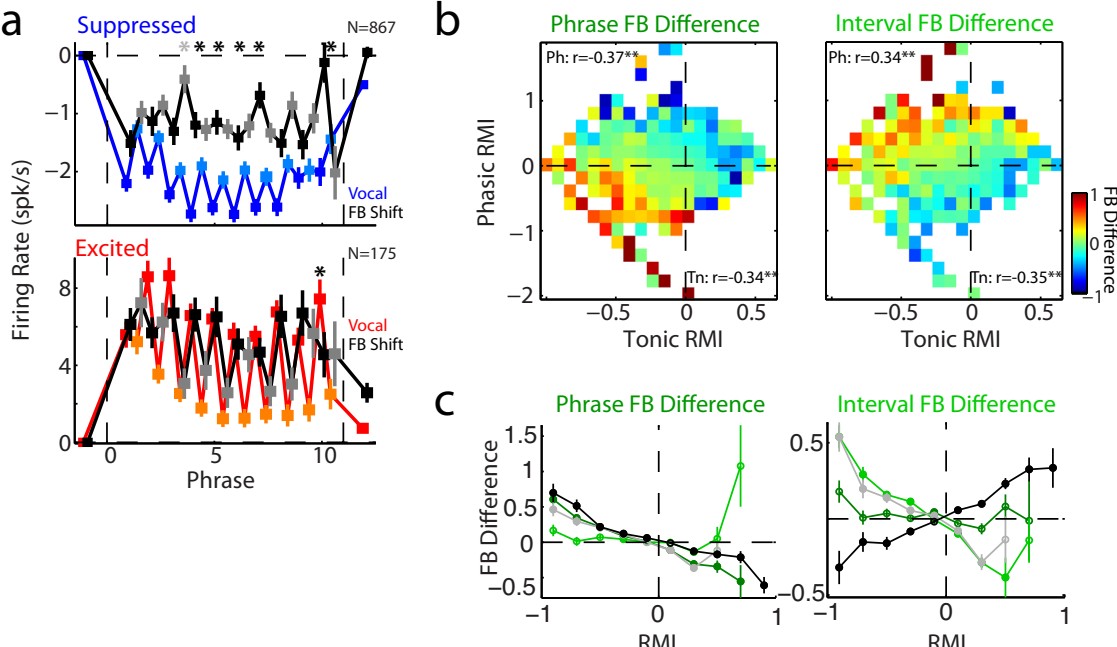

**Fig. 7 | Twitter responses during frequency-shifted feedback. a** Population mean responses for suppressed (top) and excited (bottom) units comparing firing rates during normal (colored) and shifted feedback (phrases: black, intervals: gray). Mean ± SEM are shown, and numbers of units are indicated. Suppressed units exhibited feedback increases in firing (decreased suppression) for both phrases and inter-phrase intervals, while excited units showed changes primarily during intervals. Bins with significant feedback effects are indicated (* $p < 0.05$ two-tailed signed-rank test with FDR correction, exact p-values in Source Data file). **b** Feedback-normal vocal differences are shown, binned as in other figures by normal phasic and tonic responses, and separated by feedback effects seen during phrases (left) and intervals (right). Partial correlations and significance are indicated (** $p < 0.001$; Phrase: Ph/Tn: $p = 1 \times 10^{-63}/1 \times 10^{-65}$; Interval: $p = 8 \times 10^{-65}/1 \times 10^{-68}$; partial correlations with two-sided t-tests, $n = 2366$ units). Feedback differences correlated with suppressed average/tonic responses but showed opposite relationships to phasic Phr-Int differences, correlating with phasic suppression during vocal phrases and excitation during the intervals. **c** Feedback differences, plotted separately for feedback effects seen during phrases and intervals, and binned based upon vocal RMI: phrase (dark green), interval (light green), phasic (black), and tonic (gray). Both phrases and tonic responses had a consistent relationship with feedback, with suppressed units (negative RMI) showing positive feedback effects. Interval activity had little effect on feedback during phrases but correlated with feedback effects during the inter-phrase interval (r = −0.43, $p = 6 \times 10^{-108}$, Pearson correlations with two-sided t-tests, $n = 2366$ units). In contrast, phasic activity flipped the sign of its relationship to feedback between phrases and intervals. Error bars are SEM, filled symbols indicated significant bins ($p < 0.05$, two-sided signed-rank tests with FDR correction, exact p-values in Source Data file). Source data are provided as a Source Data file.

than previously appreciated. These results may suggest that top-down motor predictions may contain information that is both general across the whole call and specific to individual phrases.

Previous work on vocal suppression in both humans and animals has largely focused on long-duration vocalizations[6,7,34], during which both tonic and phasic responses would overlap with sensory feedback, and may explain why these different components may not have been as easily distinguished before. However, their separation may explain many previously inconsistent results. For example, pre-vocal suppression has been noted in marmoset auditory cortex, yet the temporal patterns seen in PSTHs of longer vocalizations have often shown variable slopes, a gradual one before vocalization and steeper one following vocal onset[34]. We can now potentially explain this observation as the addition of phasic inhibition during vocalization to a tonic suppression beginning pre-vocally. Previous studies have also failed to detect a clear frequency-tuning specificity to vocal suppression[10,33], in contrast with vocal excitation, despite evidence for frequency-specific gamma-band inputs during vocalization[12]. Here we found that both tonic and phasic suppression were specific for neurons tuned to the vocal frequency range, although to a greater degree for the tonic component. Additionally, because individual neurons exhibited tonic and phasic responses in varying combinations, the addition of excitatory auditory inputs to an overlapping tonic vocal motor suppression of equal magnitude may explain neurons with net zero response during vocalization, which we would have previously labeled as vocally unresponsive but excited by passive vocal playback[33]. It may also be

conceivable that phasic suppression, like excitation, is also a sensory response as many auditory cortical neurons are inhibited at the loud sound levels seen during marmoset vocal production[35,36]. However, we tested playback responses at similarly loud sound levels and did not find significantly inhibited responses, making this explanation less likely.

The presence of a global or tonic vocal modulation across twitter phrases also suggests the existence of a global motor plan that similarly spans across the phrases, in addition to phrase-specific motor control. Consistent with this notion, marmoset twitters are considered to be a single vocal object, rather than simply a collection of unconnected phrases; individual twitter phrases are rarely ever produced in isolation, have predictable evolution of acoustic structure across phrases, and twitters are produced by infant marmosets with a largely complete acoustic structure[32,37]. Previous analyses of multi-phrase marmoset phee vocalizations have similarly suggested the existence of a global motor plan based upon the predictability of later phrase acoustics by earlier phrases[31]. Also consistent with global control of multi-phrase vocalizations, electrical stimulation-evoked vocalizations in a related species, squirrel monkeys, have been found to sometimes result in multi-phrase calls[38–40]. Here we examined the predictability of later twitter phrase acoustics using earlier phrases, and found correlations similar to those previously used as evidence for global motor planning in marmoset phee calls[31]. Interestingly, these correlations are strongest between adjacent phrases, which could be a result of some sort of cascading motor network. Regardless, these results suggest

that there may exist a global motor plan that spans the entire duration of the twitter call in addition to any phrase-specific motor control.

It should be noted, however, that cleanly separating out tonic and phasic response components is not trivial. While the phasic response, the difference between phrases and intervals, is intuitive, calculating the tonic response is less so. Many neurons suppressed during phrases showed interval activities well above spontaneous, possible a rebound from inhibition[41], making the interval activity a poor measure of the tonic response. Phasic Phr-Int differences were also (inversely) correlated with interval responses. The average of phrases and intervals was therefore chosen, as it maximally decorrelates from the phasic response. However, even this interpretation is difficult. For example, a positive phasic unit with a zero tonic average (i.e. Fig. 2b, example *e*) has negative/suppressed interval responses. Does this unit therefore have tonic vocal suppression, or is it reflecting a degree of adaptation after excited phrase/phasic responses? Some units with phasic excitation did exhibit interval suppression during playback, but even these showed stronger suppression during vocal production than playback. It is likely that tonic suppression is reflected in both interval as well as average responses, though either alone is an imperfect measure.

How do these results help us reconcile conflicting notions about the mechanisms and functions of corollary discharges (CD) and their role during speech and vocal production? Looking broadly across both auditory and non-auditory sensory-motor systems, CDs appear to play two general roles[25,26]. The first is a temporally-precise, but relatively non-specific in its sensory targets, inhibition aligned to movement or action onset that serves to gate or block inputs. This form of CD has been seen in cricket auditory system[27] and in rodent cortex during locomotion[42-44] or button press[45], and likely corresponds to the phasic suppression seen during marmoset twitter phrases. The second CD role is a predictive one, relaying specific predictions about the expected sensory consequences of action that alters the processing of sensory feedback. Such predictions may exhibit anticipation of an upcoming action[18] and may be more specific to the neurons or neural targets involved, as has been seen when rodent locomotion is paired with an expected sound[46]. This form of CD shares many features with the tonic modulation seen during marmoset twitters, notably pre-vocal onset, responses to vocal feedback changes, and specificity to neurons in the vocal frequency range (those most likely to exhibit a sensor consequence of vocalization in the absence of a CD). Evidence from rodent locomotion, which can exhibit both non-specific motor suppression and specific predictions, if there are learned and expected sensory outcomes to action, suggests that these two forms of CD can exist within the auditory cortex[42,46]. The present findings are consistent with such results, and extend those findings by suggesting that both CDs can co-exist even within the same neurons, mixed in different combinations and resulting in diverse vocal responses.

The presence of different types of suppressive CDs, with different degrees of gating and sensitivity to altered feedback may also explain why some human intracranial recordings have not found a clear correlation between suppressed electrodes and those sensitive to feedback[7], although other studies have noted such a correlation, particularly in primary auditory cortex[17]. Sites with a more phasic (gating CD) than tonic suppression (predictive CD), for example, might appear strongly suppressed due to the combination of both CDs, yet less responsive to altered feedback due to the variable relationship of these components with feedback sensitivity. Importantly, the current findings also extend these previous observations by demonstrating how the two processes, phasic/gating and tonic/predictive, differently affect different populations of auditory cortex neurons, with phasic responses broadly distributed across populations without frequency specificity and tonic responses being more specific for neurons in the vocal frequency range (and therefore also more likely to receive vocal feedback input). It is likely that this tonic, predictive corollary discharge process is responsible for self-monitoring and feedback-

dependent vocal control seen in both humans and marmosets[16,47]. Unfortunately, the variability of twitter vocal acoustics, and their relative sparsity, prevents ready analysis to directly address potential behavioral correlates of tonic and phasic suppression.

Interestingly, we also noted differences in these two CDs between more medial electrodes (presumed primary auditory cortex) and more lateral electrodes (lateral belt), with more predictive suppression laterally. This difference may explain why a greater dissociation of suppressed and feedback electrodes was noted in human superior temporal gyrus than in Heschl's gyrus, and may be consistent with more lateral areas having greater connectivity with frontal cortical regions[48-50]. Observed hemispheric biases between the different components, with greater tonic suppression in the right auditory cortex as opposed to similar phasic suppression between hemispheres, is also consistent with right hemispheric biases in human auditory cortex suppression[17,51] and right bias for vocal pitch control in human motor cortex[52]. This would suggest a right hemispheric bias for frequency prediction and resulting feedback-dependent control, also consistent with hemisphere bias seen in vocal control predictions and the effects of microstimulation on marmoset vocalizations[16].

In contrast to the putative behavioral role for a tonic/predictive CDs, the possible functions of the temporally precise, but non-specific, gating are a little less intuitive than for predictive inputs. While these inputs could reflect predictions about vocal acoustics beyond frequency, such as timing or loudness, they could also represent a tagging mechanism allowing a human or animal to distinguish between self-produced and outside sounds[53]. One final possibility is that non-specific suppression might represent a gating to cancel-out interference from undesirable sounds co-produced during vocalization, such as bone-conducted sounds of orofacial movement, which have been found to evoke contractions in the middle ear muscles of the peripheral auditory system[54].

The existence of multiple timescales of vocal modulation may also have other implications for human speech-motor control. Speech also operates at different timescales, including fast control of 'supra-segmental' parameters that span across multiple syllables and words, like pitch and loudness[47,55,56], and slower plastic control of shorter 'segmental' acoustics features like vowel formants that are specific to individual words[57]. While human speech occurs at a much slower rate than the phrases of marmoset twitters, it is possible that the fast and slow CD timescales observed here are also active during speech and may play different roles in segmental and supra-segmental motor control. Past work has been unable to parse out such a possibility due to the use of repeated production of single vowels or phonemes which, while controlling experimental variability, lacks the rhythmic structure of fluent speech.

Altogether, these results suggest the presence of two modulatory signals seem to co-exist to varying degrees within individual neurons in auditory cortex, rather than being segregated to different sites along the sensory hierarchy, suggesting a more complex functional and anatomic inter-dependence than has been previously suggested[25]. It is possible that tonic and phasic suppression may also have distinct origins within the vocal motor pathway, possibly including prefrontal, premotor or primary motor cortex[42,58-61]. The tonic response, for example, might reflect a direct top-down input from the frontal cortex, while phasic suppression might be a product of gating at lower areas of the sensory pathway[62-64]. Given their different temporal characteristics, it is also possible that they have distinct biological mechanisms or local neural circuitry. This distinction may also suggest possible differences in the computational roles of tonic and phasic responses, such additive vs. multiplicative gain modulation[65]. Future investigation of these processes will yield further insights into the mechanisms of top-down modulation and sensory prediction in the auditory and other sensory-motor systems.

## Methods

We recorded neural activities from five adult marmoset monkeys (*Callithrix jacchus*), one female and four males, while the animals produced self-initiated vocalizations. Neural activity from the auditory cortex, including both primary and non-primary areas, was recorded using implanted multi-electrode arrays and compared to simultaneously recorded vocal behavior. Some data were gathered as part of other experiments, where we have largely ignored the twitter calls in the past due to increased complexity of their analysis. All experiments were conducted under the guidelines and protocols approved by the University of Pennsylvania Animal Care and Use Committee and the Johns Hopkins University Animal Care and Use Committee. Three animals were studied at the University of Pennsylvania, two at Johns Hopkins. All animals had neural recordings in both hemispheres.

### Vocal recordings

Using previous methods, we recorded marmosets vocalizing while in their home colony[13,16,66]. Subjects were placed in a small cage within a custom three-walled sound attenuating booth allowing free visual and vocal interaction with the remainder of the marmoset colony. During recordings, marmosets were tethered within a small cage, to allow neural recording, but were otherwise unrestrained. Vocalizations were recorded using a directional microphone (Sennheiser ME66 or AKGC1000S) placed ~20 cm in front of the animal and digitized at 48.8 or 50 kHz sampling rate (TDT RX-8, Tucker-Davis Technologies, Alachua FL or National Instruments PCI-6052E, MATLAB controlled TDT acquisition software). Vocalizations were extracted from the recordings and spectrographically classified into established marmoset call types[32] using a semi-automated system. All major call types were produced in this context (phees, trillphees, trills, twitters), however, we primarily focused on the analysis of the marmoset twitter calls, which were produced with less frequency than the other call types. During altered feedback experiments, microphone signals were passed through a digital effects processor (Eventide Eclipse V4 or Yamaha SPX 2000) and shifted either up or down by 2 semitones, presented back to the animal at +10 dB through modified ear-bud style headphones (Sony MDR-EX10LP). Altered feedback was presented either in a blocked or random fashion, consistent with our previous work[13,16]. While animals wore the headphones throughout the duration of a vocal recording session during which altered feedback was to be tested, these did not occlude the ear canal and were not connected to any sound sources during a period of baseline 'normal' vocal production at the start of each session. Due to the high degree of acoustic variability in the first twitter phrase, we could not analyze the behavioral effects of altered feedback on vocal production as we have done in previous work.

### Neural recordings

All marmosets were implanted bilaterally with multi-electrode arrays (Warp 16, Neuralynx, Bozeman MT), one in each auditory cortex. Details of the array design and recording technique have been previously published[66]. These arrays consist of a 4 × 4 grid of individually moveable sharp microelectrodes (2–4 MΩ tungsten; FHC, Bowdoinham ME). Consistent with our previous methods, we first localized the center of primary auditory cortex using single-electrode methods, and placed arrays to cover the full range of the tonotopic axis, verified by frequency tuning. Based upon relative responses to tone and noise stimuli, electrodes were judged to likely span both primary (A1) and non-primary (belt, parabelt) auditory cortex[67]. Neural signals were observed on-line to guide electrode movement and optimize signal quality. Digitized signals were sorted off-line using custom MATLAB (R14) software and a principle component-based clustering method, and then classified as either single-unit or multi-units[66]. Due to relative scarcity of twitter calls across our recordings, low spontaneous rate of many single-units, and the short duration of twitter call phrases and inter-phrase intervals, we were concerned about a high number of units showing zero activity during the phrases/intervals. We therefore included both single- and multi-unit responses, which generally had higher firing rates, in our analysis.

### Auditory stimuli

Prior to each neural recording in the colony, we first characterized tuning properties of the auditory cortex units by the presentation of auditory stimuli. Marmosets were seated in a custom primate chair within a soundproof chamber (Industrial Acoustics, Bronx NY). Auditory stimuli were digitally generated at 97.6 kHz sampling rate and delivered using TDT hardware (System II/III) in free-field through a speaker (B&W DM601 or 686 S2) located ~1 m in front of the animal. Stimuli included tones (1–32 kHz, 10/octave; −10 to 80 dB SPL by 10 dB) and bandpass noise (1–32 kHz, 5/octave, 1 octave bandwidth), as well as wide-band noise stimuli. The center frequency (CF) of a unit's tuning was determined by the strongest response to either tone or bandpass stimuli. We further presented playback of samples of an animal's own vocalizations (previously recorded from that animal) at multiple sound levels. Although vocal stimuli were presented at multiple sound levels, only those samples overlapping vocal production loudness were used for comparisons between vocal production and auditory playback. To be consistent with our previous work, all playback sounds were presented through the speaker rather than the headphones that were used during altered feedback experiments.

### Data analysis

All analyses were performed using MALAB (R14). We calculated spontaneous-subtracted firing rates individually for each twitter phrase and inter-phrase interval. Phrases and intervals were then averaged for each produced vocalization and subsequently combined across vocalizations. For display purposes, vocal onset-aligned peristimulus time histograms (PSTHs) were calculated with 5 ms binwidth and smoothed with a 5-point moving average. Confidence intervals were calculated for PSTHs using a bootstrap method and 1000 repetitions to estimate the range containing 95% of the means. Due to call-to-call variability in the duration of individual phrases and intervals, smearing out any temporally phase-locked responses, further quantitative analyses were performed by measuring the firing rate during individual phrases and intervals, calculated call-by-call by dividing the number of spikes during each such epoch (phrase/interval) by the corresponding epoch duration, and then averaging across all phrases or intervals for that call. Consistent with previous work, we further quantified different units' responses using a normalized rate metric, the Response Modulation Index (RMI), RMI = $(FR_{voc}-FR_{pre})/(FR_{voc}+FR_{pre})$, where $FR_{voc}$ and $FR_{pre}$ are the firing rates before and during vocalization[10]. Pre-vocal firing rates were calculated from a window from 4 s to 1 s preceding vocal onset in order to exclude the effects of pre-vocal suppression which has been seen in the 200–500 msec before vocal onset[10,34]. RMI was calculated separately for phrase and interval responses, first averaging firing rates of all phrases or intervals (including the first) for a given call before normalizing with the RMI. In some analyses, units were categorized as suppressed or excited based upon a vocal RMI (comparing phrases to pre-vocal baseline) of ≤−0.2 or ≥0.1, consistent with our previous work. We further sought to determine possible tonic and phasic vocal responses that lacked the correlation of phrase and interval activity; phasic responses were defined as the phrase-interval (Phr-Int) difference, while tonic responses were calculated as the average of phrases and intervals. Similar calculations were performed for auditory playback and for vocalizations with altered feedback. Additional analysis was performed on phrase and interval RMI responses across units using a Principal Component Analysis (PCA) to determine separable patterns of activity.

Frequency tuning analyses were performed by calculating the center frequency (CF) of a unit as the frequency with the maximal firing rate response. The strongest response from either tone or bandpass noise tuning was used. Bandwidth was defined as the frequency range over which responses were at least 50% of the peak firing rate. Unit CF distance was calculated as the ratio of the unit CF relative to a reference frequency in the middle of the vocal range (7.5 kHz), in octaves. A normalized CF distance metric was also calculated by dividing the CF distance by the bandwidth. If bandwidth was larger than the CF distance (i.e. the reference frequency was within the bandwidth), the normalized distance was set as zero.

Acoustical analysis was performed on individual phrases, measuring mean frequency, loudness (SPL) and duration. Mean frequency was measured by first calculating time-binned peak frequencies from the spectrogram, and then averaging across an entire phrase duration. SPL was measured from the root-mean-squared average across the phrase duration. To exclude between-animal effects, correlations between phrases were calculated by first z-score normalizing each phrase for the mean and standard deviation of that phrase position for a given animal (i.e. all exemplars for phrase 2 for an animal were normalized using only phrase 2 data for that animal). Correlations between subsequent phrases, or groups of phrases, therefore, suggest that an acoustical variation from the mean for a given phrase predicts subsequent phrases similarly vary from the mean.

Spike phase analyses were performed to normalized spike timing within twitter phrases and intervals by designating each cycle as phrase onset to the next phrase onset. The first phrase, which is quite variable in duration, and the last phrase, which lacks a following interval, were ignored for this phase analysis. Spike times were scaled relative to each cycle duration and converted to radians. Circular analyses were performed to calculate the mean phase for each unit, and the significance for each unit (relative a flat distribution) was calculated using vector strength and circular statistics.

With the exception of correlation, regression, and phase analyses, all statistical tests were performed using non-parametric methods. Wilcoxon rank-sum and signed-rank tests (two-sided) were used to test the differences between unmatched and matched distribution medians, respectively. Kruskal-Wallis ANOVAs were used when comparing more than two conditions. Correlation values within individual units and between units were calculated with Pearson correlation coefficients. When calculating correlations simultaneously against two different variables (i.e. simultaneously against phrase and interval or tonic and phasic), a partial-correlation method was used. When binning and averaging results in a single plot, p-values were first calculated for individual bins/phrases, and then False Discovery Rate (FDR) was corrected for multiple comparisons.

### Reporting summary

Further information on research design is available in the Nature Portfolio Reporting Summary linked to this article.

## Data availability

The processed data generated in this study are provided in Source Data file. Raw data are available upon request from the corresponding author. Source data are provided with this paper.

## Code availability

Analysis code is available upon request, without restrictions, from the corresponding author.

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

## Acknowledgements

This work was supported by NIH grants DC014299, DC018525 (SJE), DC005808 (XW), funding from a Triological Society Clinician-Scientist Development Award (SJE), the Ministry of Education, Culture, Sports, Science and Technology of Japan (Leading Initiative for Excellent Young Researchers Grant 1071421 [JT], A19K237690 [JT]), The Ichiro Kanehara Foundation (JT), and Daiichi Sankyo Foundation of Life Science (JT). The authors would like to thank T. Coleman, P. Sayde, and A. Pistorio for assistance with animal training and care, and Richard Mooney for comments on this manuscript.

## Author contributions

J.T., X.W. and S.J.E. contributed to the design of the experiments. J.T. and S.J.E. collected the data and performed the analysis. J.T., X.W. and S.J.E. wrote and edited the manuscript.

## Competing interests

The authors declare no competing interests.
