## [Peer Review File · Nature Communications]

Multiple processes of vocal sensory-motor interaction in primate auditory cortexREVIEWER COMMENTS

Reviewer #1 (Remarks to the Author):

In this study Tsunada and colleagues investigated the presence of different vocal suppression mechanisms in the auditory cortex of marmoset monkeys during vocal production. The authors performed electrophysiological recordings in freely moving and vocalizing monkeys and analyzed the responses of single neurons in the auditory cortex during twitter call production. The present work highlights the presence of two distinct components of vocal suppression: a phasic one, that occurs during individual vocal phrases, and a tonic one that begins before vocal onset and persists for the entire call duration. By playing twitter vocalizations to the monkeys through earphones, they show that neurons that were suppressed during vocal output did not show any significant suppression during playback, suggesting that suppression was specific to self-produced vocalizations rather than auditory feedback. Overall, the results of this study indicate that there are two timescales of vocal suppression which may reflect different mechanisms of vocal self-monitoring and vocal motor control.

This interesting work nicely builds on the earlier work of the PI of the present paper and provides new and important insights into the neural suppression mechanisms occurring during vocal output. The study is well written, methodologically sound and the number of animals used is sufficient for a study with non-human primates. I have only a few minor comments. This work will be of great interest and will be suitable for publication in Nature Communication once my concerns below have been sufficiently addressed.

1. Introduction: The authors could add some brief information about the methodological approach at the end of the introduction. This will help to better understand the results without jumping back and forth to the methods section. For example, mentioning that earphones were used for playback would make it easier to access the results section and the playback plots on Figure 1.
2. Line 114 and Figure 1D: The plot shows an increase in activity during the early inter-phrase interval – could the authors provide an explanation for this phenomenon? Is there a possible implication?
3. Line 120: The authors should provide information on how the baseline neural activity was calculated.
4. Line 133: The phrase-interval difference is reported as “P-I difference” in the text, while it seems to be reported as “Phr-Int Difference” in the Figures (e.g. 3B). Please adjust accordingly.

5. Line 150ff: Please provide more details about the earphones manipulation. It is unclear whether the animals wore the earphones only during frequency shifting and playback, or whether they wore them throughout the whole data collection (and if so, whether the twitter calls produced by the animals were also played back without frequency shifting). Where the twitter calls used during the playback experiment from the animal itself or another one?

6. Line 423ff: Please provide more information on how exactly the P-I difference RMI was calculated. Was the calculation performed by taking the means of all phrases and the means of all intervals and the averaging over all calls, or by calculating the difference between each "segment" (phrase and associated interval) and then averaging for all "segments"? Please clarify.

7. Figure 3B: The different shades of green and blue are difficult to distinguish and should be improved.

Reviewer #2 (Remarks to the Author):

This paper describes differential responses to auditory cortical suppression during vocal production when compared to responses to playback. Both suppressed and excited units show a differential response to phrase and inter-phrase periods during a twitter call during vocal production which is not observed during playback. There appears to be greater modulation between phrase and inter-phrase responses in excited units rather than suppressed units. The functional significance of this difference however is quite unclear. Overall the study describes an intriguing finding that A1 neurons show both tonic and phasic suppression to twitter vocalizations and this suggests two types of suppressive mechanisms. Since twitter calls are made without breaks or subcomponents, the authors posit that they should have a single motor program. While that were possible, it is also possible that the motor program includes dynamics that reflect the phrase structure of the call. If that is the case the data presented could represent the complex nature of the predictive signal that contributes to cortical suppression during vocal production. It could also be that there are signals that could be used to convey the completion or execution of the twitter call in terms of the number of phrases. But this issue is not at all addressed in the paper.

While there is a nugget of an interesting finding, the paper is unfortunately quite poorly written. The figures are complicated and described quite poorly, and the author assumes self-evidence in the figures which is far from present in many cases. Overall the computational significance of this work is unclear in terms of models of vocal production and control. The behavioral significance is also unclear. Also, in humans there is similar heterogeneity in suppressive and feedback-sensitive responses that are not discussed in relation to the current findings.

Specific Comments:

Title: It is not clear what multiple mechanisms refers to in the title.

Abstract: A bit more needs to be said about distinct sensory modulation in other domains at multiple levels. Overlap of suppression with sensory feedback is a bit unclear. The suppression is observed in the presence of sensory feedback. Why is this a bottleneck for determining mechanisms of sensory-motor integration or presence of detection or characterization of multiple top-down signals? The problem is not well set up.

Maybe the real question that is being posed here is the time-scale of suppression and whether there is any dynamics to the suppression signals.

What is it mean by distinct sensory properties?

What are putative differing mechanisms or functions?

Introduction:

The first paragraph is a confusing rendition about two phenomena observed in auditory cortex during vocal production - suppression of sensory feedback at onset, and enhancement of responses to feedback perturbations that their putative functional roles.

The word mechanism is used interchangeably to describe a phenomenon, its neural basis and its functional significance. Please clarify.

The production of multiple twitters alone does not constitute evidence for a single motor program.

A bit more needs to be stated about the model or framework for hypothesized mechanisms for whole-vocalization vs phrase-level suppression. What are principles that may guide different forms of suppression? What would models predict - both?

Results:

The intro sets up examination of tonic vs phasic responses but the first few figures do not address this question and look at phrase vs interval.

The time-warping procedure is poorly motivated and justified. The conclusion of phrase level suppression for this neuron is apparent in figure 1C itself. It will be useful to include the gray regions corresponding to the phrases here in this subplot. It may help to show the time-warping results for different vocalizations from the same animal and also from different animals to be able to discern patterns that are revealed better with warping.

Figure 1F-H are poorly described in the results section. The procedure for excited vs suppressed units definition needs to be made clear. Specifically are excited vs suppressed based on a comparison of vocal production vs playback, playback alone or vocal production responses alone? The statistics seem dodgy here without consideration of either time-series data or multiple comparisons issues. It doesn't mention that G is a time-warped version of F is that so?

A simpler analysis could be one looking at phrase-level response and inter-phrase responses pooled across animals and vocalizations. Similar to figure 3A but for all units.

Figure 1H is quite confusingly plotted and phrased. Is the Vocal RMI referring to suppressed or excited units? What is the color scale being plotted then? There is no legend for what the color corresponds to? What is a middle response unit?

It appears that the RMI corresponds to the response to vocal productions. The color scale corresponds to the degree of suppression or enhancement? But this is a source of huge confusion for the reader. It is far from obvious.

Are phrase level RMI computed by averaging across all phrases? Is the first phrase which would correspond to the onset excluded? Could there be interesting dynamics in the phrase level response modulation?

Figure 2 is also confusing in multiple ways. The non-significant orange circles are only outside the data range in the large suppression. What does it mean to say a bias for less interval suppression? What is being plotted in the top inset? Again, not self-evident.

What is the point of Figure 2B? Inferences from this analyses do not seem rigorous.

Essentially what seems to be show is that there is only a moderate correlation between phrase vs interval modulation. So, they seem largely separable. Could the moderate correlation be seen because of some of artifacts arising from the analyses such as interval responses being included in the phasic responses.

If units are separated based on whether they have phasic vs onset responses do their suppression for vocal production merely reflect their temporal responses properties?

Figure 3A could first show all units. The excited units colors are confusing in terms of red and orange correspond also to playback? There is no indication.

It appears that the excited units show less inter-phrase suppression more than anything else. Why is this not seen in figure 1H?

Figure 3C is uninterpretable. What is the color scale here? Figure 3B suggests that there is no difference in the phrase vs interval responses during playback but this is seen in vocal production, particularly for the excited subpopulation.

Perhaps the correlation between playback interval vs vocal interval responses can be looked at.

The results seem to mixed with interpretative discussion such as "... could be explained by background noise in playback stimuli ..." which are difficult to understand.

While this is observed during vocal production "...tonic/interval suppression during vocal production combined with a phasic sensory excitation ..." why does this not occur during playback?

What is missing in the frequency tuning based analysis is an analysis of the temporal response phase with respect to the twitter phrase? Are the phases preserved during vocal production vs playback?

It is only until figure 4 that there is an analysis of the middle units. Why is that?

Why is the phasic RMI as a function of CF only shown for the suppressed units? What is being plotted in figure 4D?

What is different in the plots in figure 4E compared to the difference plots in figure 4C?

The colors in figure 5 correspond to what? Unclear.

Figure 5B is also impossible to follow. If the goal is to show the correlations for tonic but not phasic, then a plot similar to figure 5A may be warranted. The point of figure 5B is unclear and is confusing.

It seems this analysis should also be split across suppressed, middle and excited units but that seems to be missing in the text although briefly mentioned in the text.

It is difficult to also interpret the shifted feedback results which show generalized reductions in suppression not related to phrase vs intervals.

There is a disconnect between the text in lines 238-250 and the figure itself.

There is a complete absence of any spatial analysis or bandwidth of unit related analyses or relationship to their temporal response or modulation properties.

There are some discrepancies between figure 1G and figure 3A & B. Is that due to time-warping in the former? Was subsequent analyses done before or after time-warping?

Discussion

Unclear how "combination of phasic excitation from auditory inputs and tonic motor suppression may explain neurons unresponsive during vocalization but showing strong excitation during passive vocal playback".

The inhibition hypothesis is already ruled out because the phasic responses are only seen in vocal production and not in playback.

The definitions of EC vs CD are not consistent with more recent literature and reviews on this topic. In fact in lines 316-20 the authors it would make more sense to flip EC and CD. Efference copy may be more consistent with a phasic response and CD may reflect some higher order processes that could result in either tonic or phasic responses.

The last paragraph is highly speculative and well beyond the scope of this paper, may consider toning this speculation down.

Methods: It may be useful to know how many animals data were collected at Hopkins vs Penn and if there were any differences. How many in each hemisphere by site?

In line 424 is there a typo for the formula for RMI?

We thank the reviewer for their detailed and insightful comments. In response to these comments, we have significantly revised the manuscript, and added additional analyses. We apologize for the long interval in this response, many of the suggested analyses required us to go back to re-examine primary data, which took some time. We detail our point-by-point reply below. The reviewer comments have been italicized, our responses colored in blue, and our edits to the original manuscript have been colored in red. Additionally, we decided to move the analysis of anatomic location from the Extended Data to the primary figures.

Reviewer #1 (Remarks to the Author):

In this study Tsunada and colleagues investigated the presence of different vocal suppression mechanisms in the auditory cortex of marmoset monkeys during vocal production. The authors performed electrophysiological recordings in freely moving and vocalizing monkeys and analyzed the responses of single neurons in the auditory cortex during twitter call production. The present work highlights the presence of two distinct components of vocal suppression: a phasic one, that occurs during individual vocal phrases, and a tonic one that begins before vocal onset and persists for the entire call duration. By playing twitter vocalizations to the monkeys through earphones, they show that neurons that were suppressed during vocal output did not show any significant suppression during playback, suggesting that suppression was specific to self-produced vocalizations rather than auditory feedback. Overall, the results of this study indicate that there are two timescales of vocal suppression which may reflect different mechanisms of vocal self-monitoring and vocal motor control.

This interesting work nicely builds on the earlier work of the PI of the present paper and provides new and important insights into the neural suppression mechanisms occurring during vocal output. The study is well written, methodologically sound and the number of animals used is sufficient for a study with non-human primates. I have only a few minor comments. This work will be of great interest and will be suitable for publication in Nature Communication once my concerns below have been sufficiently addressed.

We thank the reviewer for their strong support for this work.

1. Introduction: The authors could add some brief information about the methodological approach at the end of the introduction. This will help to better understand the results without jumping back and forth to the methods section. For example, mentioning that earphones were used for playback would make it easier to access the results section and the playback plots on Figure 1.

We thank the reviewer for the suggestion. We have added a brief overview of the methods at the end of the introduction detailing the use of playback and frequency shifted feedback with headphones. (page 5)

2. Line 114 and Figure 1D: The plot shows an increase in activity during the early inter-phrase interval – could the authors provide an explanation for this phenomenon? Is there a possible implication?

The reviewer raises makes an important observation and interesting point. Several possibilities exist including rebound from inhibition. We have added some discussion on this point to the results, particularly in context of Figure 2A where several units were seen in the upper left quadrant, suggesting some suppressed units with interval activities above spontaneous (page 7-8), and also in the discussion on page 19.

3. Line 120: The authors should provide information on how the baseline neural activity was calculated.

Baseline (pre-vocal) activity was calculated from a window from 4 to 1 second preceding oval onset (to exclude known pre-vocal suppression). We have added this information to the methods (page 25).

4. Line 133: The phrase-interval difference is reported as “P-I difference” in the text, while it seems to be reported as “Phr-Int Difference” in the Figures (e.g. 3B). Please adjust accordingly.

We thank the reviewer for the suggestion. We have adjusted the text to match the figures (Phr-Int)

5. Line 150ff: Please provide more details about the earphones manipulation. It is unclear whether the animals wore the earphones only during frequency shifting and playback, or whether they wore them throughout the whole data collection (and if so, whether the twitter calls produced by the animals were also played back without frequency shifting). Where the twitter calls used during the playback experiment from the animal itself or another one?

We apologize for the ambiguity. As in all our past experiments, the animal wore the headphones throughout the vocal production experiment, even if they were not being used. This was necessary as to not require re-catching the animal. These do not occlude the ear canal and for baseline/normal vocalizations without feedback these were not connected to the feedback apparatus. This is consistent with our previous work. We have added this information to the methods (page 23). The calls used during the playback experiment were the animal's own vocalizations, we have clarified in the methods (page 24).

6. Line 423ff: Please provide more information on how exactly the P-I difference RMI was calculated. Was the calculation performed by taking the means of all phrases and the means of all intervals and the averaging over all calls, or by calculating the difference between each "segment" (phrase and associated interval) and then averaging for all "segments"? Please clarify.

We apologize for the ambiguity. We first calculate the firing rate during each individual phrase and interval (for each call), and then averaged all the phrases and intervals for the entire call, yielding 2 numbers per call (phrase and interval), calculated the RMIs, we then averaged the phrase and interval RMIs across all the calls. We have clarified in the methods (page 25).

7. Figure 3B: The different shades of green and blue are difficult to distinguish and should be improved.

We have changed the shading of the blues and greens throughout to try to better differentiate.

Reviewer #2 (Remarks to the Author):

This paper describes differential responses to auditory cortical suppression during vocal production when compared to responses to playback. Both suppressed and excited units show a differential response to phrase and inter-phrase periods during a twitter call during vocal production which is not observed during playback. There appears to be greater modulation between phrase and inter-phrase responses in excited units rather than suppressed units. The functional significance of this difference however is quite unclear. Overall the study describes an intriguing finding that AI neurons show both tonic and phasic suppression to twitter vocalizations and this suggest two types of suppressive mechanisms. Since twitter calls are made without breaks or subcomponents, the authors posit that they should have a single motor program. While that were possible, it is also possible that the motor program includes dynamics that reflect the phrase structure of the call. If that is the case the data presented could represent

the complex nature of the predictive signal that contributes to cortical suppression during vocal production. It could also be that there are signals that could be used to convey the completion or execution of the twitter call in terms of the number of phrases. But this issue is not at all addressed in the paper.

We thank the reviewer for their interest in our work, and the intriguing possibility of a more complex sensory prediction. In response to these comments, and others below, we have also suggested that the motor plan being relayed may include both information about the whole call as well as individual phrases (introduction: page 5, discussion page 17).

While there is a nugget of an interesting finding, the paper is unfortunately quite poorly written. The figures are complicated and described quite poorly, and the author assumes self-evidence in the figures which is far from present in many cases. Overall the computational significance of this work is unclear in terms of models of vocal production and control. The behavioral significance is also unclear. Also, in humans there is similar heterogeneity in suppressive and feedback-sensitive responses that are not discussed in relation to the current findings.

We apologize for the lack of clarity in the writing and figure descriptions. We fully recognize that many of our figures were quite detailed and complex. We have significantly revised the manuscript and labored to clarify both the manuscript and figure legends, as well as simplify the figures where we could. We have expanded our discussion of the possible behavioral significance in terms of the relative needs to attenuate self-generated sounds (gating) vs make specific predictions for vocal control (page 20-21), but also acknowledge that it may be possible that both are needed for vocal self-monitoring. Unfortunately, the dynamic acoustics of twitters make it difficult to directly look for behavioral correlates as we have done in past work. Our hope is that we can build upon the current findings/hypothesis to target these potentially different processes and determine their relative roles in behavior.

The reviewer also raises an important point about the comparison to heterogenous findings seen in human intracranial findings (notable Eddie Chang's paper). We have briefly discussed this in our original manuscript discussion, but have significantly expanded this point in the introduction (page 4) and discussion (page 18)

Specific Comments:

Title: It is not clear what multiple mechanisms refers to in the title.

We apologize for the confusion on this point. We have retitled to ‘multiple processes.’ While ‘mechanism’ is often used loosely by neurophysiologists to describe neural correlates of behavior or other higher-level phenomena, to someone looking at finer details, like synaptic transmission, the term mechanism might be overly broad.

Abstract: A bit more needs to be said about distinct sensory modulation in other domains at multiple levels. Overlap of suppression with sensory feedback is a bit unclear. The suppression is observed in the presence of sensory feedback. Why is this a bottleneck for determining mechanisms of sensory-motor integration or presence of detection or characterization of multiple top-down signals? The problem is not well set up.

Maybe the real question that is being posed here is the time-scale of suppression and whether there is any dynamics to the suppression signals.

We apologize for the ambiguity. We have tried to clarify that the overlapping motor and sensory signals makes it difficult to disentangle the sensory-motor integration (as there is never a motor signal without sensory inputs), whether there could be multiple processes or mechanisms as seen in other sensory-motor systems, and whether there are temporal dynamics (page 2)

What is it mean by distinct sensory properties?

We have clarified to indicate that these components had different correlations with sensory tuning (page 2).

What are putative differing mechanisms or functions?

We are trying to suggest that the existence of two motor inputs may suggest that the different inputs may have different neural mechanisms and roles in behavior (as we will later address in the discussion)

Introduction:

The first paragraph is a confusing rendition about two phenomena observed in auditory cortex

during vocal production - suppression of sensory feedback at onset, and enhancement of responses to feedback perturbations that their putative functional roles.

We apologize for the confusion here. Our goal is to briefly introduce the two key neural processes at work, suppression and feedback sensitivity. We have tried to clarify, and also note that the relationship between the two is still unclear (despite the correlation in our previous marmoset work) (page 3).

The word mechanism is used interchangeably to describe a phenomenon, its neural basis and its functional significance. Please clarify.

The reviewer astutely points out that the interpretation of the term mechanism is relative, both a neural process or response can be a mechanism for a behavior, and a circuit can be a mechanism for a neural phenomenon. Most authors use this term rather loosely. However, the point is well taken, and we have tried to limit our use and be more specific when using the term. We have removed the term mechanism from our title.

The production of multiple twitters alone does not constitute evidence for a single motor program.

The reviewer raises a good point. The point that we are trying to relay is that the twitter is not just a series of completely independent elements. We have revised our statement to suggest that twitters production may involve both a global and a phrase-specific motor plan. We also point out evidence based on lack of independent phrase production, predictable acoustic evolution across phrases, and wholistic production by infants (page 5)

A bit more needs to be stated about the model or framework for hypothesized mechanisms for whole-vocalization vs phrase-level suppression. What are principles that may guide different forms of suppression? What would models predict - both?

We thank the reviewer for helping us better clarify the working model we are setting up for this paper. We have expanded our introduction about weighing what we might expect in terms of the 'lower' and 'higher' corollary discharge models, noting: "A 'lower' corollary discharge, for example, might be expected to only modulate neural activity during phrases of a multi-segmental vocal event, but be non-specific on which neurons it targets, while a 'higher' corollary discharge

might be expected to be more specific about which neurons it targets, and be result in stronger feedback sensitivity” (page 4)

Results:

The intro sets up examination of tonic vs phasic responses but the first few figures do not address this question and look at phrase vs interval.

We have clarified this in the introduction, noting how the tonic/phasic (which are likely the more important result) relate to phrases and intervals, which are the most directly observable (page 6). We then go on to discuss the transition in Figure 2 (page 7).

The time-warping procedure is poorly motivated and justified. The conclusion of phrase level suppression for this neuron is apparent in figure 1C itself. It will be useful to include the gray regions corresponding to the phrases here in this subplot. It may help to show the time-warping results for different vocalizations from the same animal and also from different animals to be able to discern patterns that are revealed better with warping.

We thank the reviewer for the suggestion, which addresses an important point that needs to be clarified. While Fig 1C (the example) as well as the population PSTH (now Fig 1E) suggest a phasic response, quantifying this cannot be done based upon the onset-aligned responses alone. As one can see in the raster (Fig 1B), there is considerable variability in the phrase durations and timing (shaded), a result of the natural variability of marmoset vocalizations. To better demonstrate this variability, we have added a histogram below the two PSTHs in Figure 1 (1C and 1E) to give a sense of the variability and the rough timing of phrases (discussed on Page 6). Additionally, we have removed the time warping entirely, as it does not seem to add much in light of the reviewer’s comments. Instead we just show the firing rates for the phrases and intervals in Fig 1D.

Figure 1F-H are poorly described in the results section. The procedure for excited vs suppressed units definition needs to made clear. Specifically are excited vs suppressed based on a comparison of vocal production vs playback, playback alone or vocal production responses alone?

We apologize for the confusion, while these details were present in the Methods, we recognize that many readers will skip over the methods section. We have clarified in the main text that suppression is defined based on RMI, comparing vocal to pre-vocal firing (page 6), which is how we have defined it in all of our past papers.

The statistics seem dodgy here without consideration of either time-series data or multiple comparisons issues.

The method we originally used, calculating p-values for individual bins and then correcting for multiple comparisons (using an FDR method) is a common approach that we and many others have previously used. Nonetheless, the reviewer's point is well taken. As the proper way to calculate significance of PSTHs is beyond the scope of this manuscript, and is not critical to the results, we have removed the calculation. Instead, we have changed our error bars in Fig 1C and 1E to be 95% bootstrapped confidence intervals, and will leave it to the readers to make their own inferences.

It doesn't mention that G is a time-warped version F is that so?

This was correct, however, we have removed the time-warping from the manuscript.

A simpler analysis could be one looking at phrase-level response and inter-phrase responses pooled across animals and vocalizations. Similar to figure 3A but for all units.

We thank the reviewer for the suggestion. As requested, we have removed the time-warping and just plot the mean phrase and interval responses for all units (Fig 1F), which shows overall suppression with phrase-interval differences.

Figure 1H is quite confusingly plotted and phrased. Is the Vocal RMI referring to suppressed or excited units? What is the color scale being plotted then? There is no legend for what the color correspond to? What is a middle response unit?

It appears that the RMI corresponds to the response to vocal productions. The color scale corresponds to the degree of suppression or enhancement? But this is a source of huge confusion for the reader. It is far from obvious.

We apologize for the confusion. We have tried simplify this plot (now Fig 1G) to show the full data from the entire neural population. We now plot the phrase and interval responses for each unit, aligned from most suppressed to most excited (based on firing rate instead of RMI), and indicate the color scale as the firing rate. Because individually units were noisy, we did end up averaging the units into blocks of 50 to make it easier to visualize. This plot shows that for

suppressed units at the bottom, phrase responses are more negative than intervals, while the opposite is true for the excited units near the top (page 6).

Are phrase level RMI computed by averaging across all phrases? Is the first phrase which would correspond to the onset excluded? Could there be interesting dynamics in the phrase level response modulation?

All phrases were included in the averaging, including the first phrase, we have emphasized this in the methods (page 25). We did some additional analysis to compare the first phrase to the subsequent ones, and found a strong correlation (i.e. suppression wasn't an onset response during the first phrase followed by adaptation/depression after, page 8). In our original extended data figure 2, we performed a PCA analysis which did show temporal patterns including a decrease over phrases (PC2), and increase over time (PC3), and onset-like response (PC4), though these had low explanatory power (all less than 7%). The significance of these is unclear, as they were equally balanced in numbers. We did some exploratory analyses but did not find that PC2/PC3 (decreasing and increasing dynamics) clearly correlated with any other variables. We note the lack of clear significance in the temporal evolution in the revised manuscript (page 8).

Figure 2 is also confusing in multiple ways. The non-significant orange circles are only outside the data range in the large suppression. What does it mean to say a bias for less interval suppression? What is being plotted in the top inset? Again, not self evident.

We apologize for the confusion here. The orange line indicates a local average (based upon bins of the phrase response) to give a sense of how the interval responses vary with the phrase. We probably could have done a linear regression plot instead, but were concerned about possible non-linearities here (though were not evident). The non-significant orange bins on the right are for units with excited phrases, but had very small sample sizes. The point we are trying to make is that, in general the intervals have the same tendency as the phrases but less strongly so than what was seen in the phrases. We have tried to clarify this (page 7). The top inset is the Phrase-interval difference distribution, we moved this to Fig 2B to be more consistent.

What is the point of Figure 2B? Inferences from this analyses do not seem rigorous.

Figure 2B (now Fig 2C) is plotted to direct show our phasic-tonic comparison, although somewhat redundant, we feel this is important to show the data directly as many subsequent figures are plotted in a similar tonic vs phasic projection. We have added some example units in Fig 2B, and labeled their corresponding locations in 2A and 2C to illustrate the diversity of

responses seen in the population, and to give the reader a sense of what it means to be a unit in a given location in the raster plot.

Essentially what seems to be show is that there is only a moderate correlation between phrase vs interval modulation. So, they seem largely separable. Could the moderate correlation be seen because of some of artifacts arising from the analyses such as interval responses being included in the phasic responses.

We would not interpret this as clearly separable. While the degree of correlation was moderate in an absolute sense ($r=0.46$), this is actually quite high for a study comparing properties of individual units/neurons, which can be very, very noisy. We have published primary outcomes in the past where the correlations were less than 0.4, and were considered strong outcomes. There is certainly a possibility of bleeding over of phrase activity into the intervals, which we have added a discussion of (pg 7-8), including the fact that phasic responses also correlate with intervals (also in the range of $r \sim 0.45$). This is the reason we felt it important to choose metrics that minimized the correlations. We also discuss the difficulty of interpreting tonic responses in the discussion, i.e. is tonic activity best represented using our phrase-interval average, interval only, or both (page 19). The PCA analysis we performed (Extended Data Fig 2, page 8) would also seem to argue in favor of the tonic and phasic approach being the best separability.

If units are separated based on whether they have phasic vs onset responses do their suppression for vocal production merely reflect their temporal responses properties?

This is an excellent suggestion, we examined onset activity and found that it correlated strongly with both phrase and interval responses (page 8). Interestingly the PCA analysis suggested that onset activity (PC4) could be separated from both tonic (PC1) and phasic activity (PC2/3), though PCA inherently results in orthogonal/decorrelated projection.

Figure 3A could first show all units. The excited units colors are confusing in terms of red and orange correspond also to playback? There is no indication.

We apologize for the confusion. We have added labels to indicate the black/grey is playback, and have emphasizes this in the legend. Instead of showing all units in 3A, we decided to add the middle group instead, since this is brought up in a later comment.

It appears that the excited units show less inter-phrase suppression more than anything else. Why is this not seen in figure 1H?

The reviewer is correct, that these excited units show strong phrase responses, but weak interval activity without suppression, in contrast to the decreased overall activity in suppressed units. The apparent may have been an artifact of the original way we had plotted Fig 1H (which has not been removed). The new 1G shows responses that are more consistent with Fig 3A.

Figure 3C is uninterpretable. What is the color scale here?

We apologize for the confusion. We have added label to the color bar to indicate that it is the Playback RMI, and we have expanded the figure legend discussion for Fig 3C to better describe how this is showing the average playback responses as a function of binned vocal phasic and tonic activity.

Figure 3B suggests that there is no difference in the phrase vs interval responses during playback but this is seen in vocal production, particularly for the excited subpopulation.

During our re-analysis addressing other suggestions from the reviewer, we found a calculation error that blurred the phrase-interval timings for a subset of the playback stimuli. This may explain while we originally didn't see much phrase-interval difference during playback. After correcting this, we now see a small difference between playback phrases and intervals (Fig 3B). This is still small owing to the suppressed and middle neurons which show small differences between playback phrases and intervals, compared to excited (where we see it for both playback and vocalization).

Perhaps the correlation between playback interval vs vocal interval responses can be looked at.

This is an excellent suggestion that we had included in our original Extended Data Figure 3. We saw that interval responses were reduced during vocalization compared to playback for most units (Ext Data Fig 3G). We have added a little more text in the main manuscript about this difference, and how it suggests tonic suppression (page 10).

The results seem to mixed with interpretative discussion such as "... could be explained by background noise in playback stimuli ..." which are difficult to understand.

We have removed this confusing statement.

While this is observed during vocal production "...tonic/interval suppression during vocal production combined with a phasic sensory excitation ..." why does this not occur during playback?

We have clarified this statement (page 10). These middle units had excitatory playback responses, but would not have had the vocal suppression added in during playback alone. The net result (auditory excitation + vocal suppression) is something that appears unresponsive during vocalization.

What is missing in the frequency tuning based analysis is an analysis of the temporal response phase with respect to the twitter phrase? Are the phases preserved during vocal production vs playback?

We thank the review for this excellent suggestion. We have performed an analysis of the phase of spiking during vocal production and playback using circular statistics (Ext Data Fig 5, page 12-13). We found that during vocal production most spikes occurred during the interval (particularly for suppressed units), while during playback and for excited units, spikes were more likely to occur during phrases. When we looked at mean phase vs unit CF, we found a CF-dependent phase progression in playback/excited units that was similar to that seen in Xiaoqin Wang's 1995 J Neurophys paper. We did not see any phase dependence for the suppressed and negatively phasic units.

It is only until figure 4 that there is an analysis of the middle units. Why is that?

This is an oversight on our part, we have added middle units to Fig 3 as well, and also discuss unresponsive/middle units in the context of the Fig 1G and the scatter plots of Fig 2.

Why is the phasic RMI as a function of CF only shown for the suppressed units? What is being plotted in figure 4D?

What is different in the plots in figure 4E compared to the difference plots in figure 4C?

We apologize about the confusion on our original 4D + 4E. We were attempting to find a way to determine if there was a difference in the frequency specificity for phasic suppression. While interval/tonic suppression appeared specific to vocal frequency units, and tonic excitation/playback similarly frequency-dependent, it was hard to pull apart the frequency specificity for phasic suppression. We have instead added a series of analyses to better try and address this point. In the new Fig 4D, we compare phrase, interval, difference, and playback responses for all units to the middle of the vocal range (7.5 kHz), and show statistics indicating their tuning (pg 11-12). We also compare CF distance (relative to 7.5k) vs tonic and phasic responses in the new Fig 4E. Finally, we also compare this CF distances for playback responses. These new analyses show that units with phasic and playback excitation have CFs closer to the vocal range than phasic/playback suppression. Interval and tonic responses, on the other hand, show closer CFs for many suppressed units. These results suggest that interval/tonic suppression, as opposed to phasic suppression, is seen in units whose CFs more closely overlap the vocal range.

The colors in figure 5 correspond to what? Unclear.

Figure 5B is also impossible to follow. If the goal is to show the correlations for tonic but not phasic, then a plot similar to figure 5A may be warranted. The point of figure 5B is unclear and is confusing.

We thank the reviewer for the suggestion. We have replotted the old Fig 5B to be more similar to 5A (a scatter plot). We also moved the old Extended data Figure comparing twitters to other calls to this figure (Now Fig 5C).

It is difficult to also interpret the shifted feedback results which show generalized reductions in suppression not related to phrase vs intervals.

We agree that this is one of the more perplexing findings in these results. If the feedback sensitivity is a result of the tonic suppressive component, as we are proposing, then this input is likely present during both phrases and intervals, we have expanded a discussion of this point (page 20). We present some alternative possibilities, and control analyses. We speculate that because interval feedback effects were seen primarily for units with phasic excitation (which appears to be an auditory response), the presence of delays from the feedback hardware placing novel sound energy in the intervals could account for the observed effects. Unfortunately the playback controls we tested did not quite match the degree of hardware delay seen during vocal production. We did attempt to exclude the early interval period in an additional analysis, but the results were difficult to interpret. We also did a control analysis looking at bias from floor effects (i.e. very low firing rates that could not go any lower anyway), but saw similar patterns whether units had a high or low spontaneous rate (Extended Data Fig 7, page 16).

There is a disconnect between the text in lines 238-250 and the figure itself.

We again apologize for the confusion, we have attempted to clarify the text (page 15), and have replotted Fig 7C in what we hope is more accessible plot. Our main point is that we see feedback effects generally more for suppressed than excited units (Fig 7A), and that there is a consistent correlation of feedback changes with tonic suppression (i.e. stronger feedback increases with larger tonic suppression). Perplexingly, we also see feedback effects that correlated with phasic activity but this seems to change from phrases and intervals, which we cannot fully explain but discuss above.

There is a complete absence of any spatial analysis or bandwidth of unit related analyses or relationship to their temporal response or modulation properties.

This is an excellent suggestion, and we have added a band-width based analysis (Extended data figure 4, page 11). We found that excitatory units had larger bandwidths, as did units with stronger playback responses, both of which are unsurprising. We also compared vocal/playback responses as a function of CF distance from our reference of 7.5 kHz (as in Fig 4), but now normalized by bandwidth, which slightly improved the frequency dependence of the responses.

Unfortunately, aside from the phase analysis the reviewer suggested, we don't really have much additional data on temporal modulation dependence. In order to collect sufficient vocal production, we had to be selective with the auditory stimuli we could present in the same session, and so could not explicitly test temporal modulation responses. We presume these properties would be reflected in the playback responses, however, which we did examine. We had the similar practical limitation to performing any analysis of spatial position tuning (though previous studies on marmoset auditory cortex spatial tuning have shown some effects on response magnitudes, it rarely results in inhibition below spontaneous).

There are some discrepancies between figure 1G and figure 3A & B. Is that due to time-warping in the former? Was subsequent analyses done before or after time-warping?

The reviewer is correct, these probably reflect the effects of the time-warping which can sometimes distort firing rates when there is large variance in the time intervals (as we saw). We have removed the time warping due to these concerns.

Discussion

Unclear how "combination of phasic excitation from auditory inputs and tonic motor suppression may explain neurons unresponsive during vocalization but showing strong excitation during passive vocal playback".

We are sorry for the unclear statement. We have revised it to note that a linear addition of excitation from auditory inputs to an overlapping tonic vocal suppression could result in a net zero response that we would have classified as unresponsive in our previous work. (page 18)

The inhibition hypothesis is already ruled out because the phasic responses are only seen in vocal production and not in playback.

While we attend to agree, this is an issue that has been raised multiple times by reviewers in our past manuscripts, and so we feel it important to clearly state that these responses cannot be explained by playback inhibition, though a few units did show some interval suppression during

playback, but not to the degree seen during vocal production. We have added a discussion on this point (page 19).

The definitions of EC vs CD are not consistent with more recent literature and reviews on this topic. In fact in lines 316-20 the authors it would make more sense to flip EC and CD. Efference copy may be more consistent with a phasic response and CD may reflect some higher order processes that could result in either tonic or phasic responses.

The specific definitions and use of the terms EC vs CD tend to be inconsistent in the literature, something we are also quite guilty of. To avoid this debate, we have reframed the discussion point in terms of the previously proposed ‘lower’ and ‘higher’ corollary discharge, discussing them instead as non-specific/gating vs predicting inputs (page 20).

The last paragraph is highly speculative and well beyond the scope of this paper, may consider toning this speculation down.

We thank the reviewer for the suggestion. We have toned down the discussion to speculate they may have distinct origins in the pathway, and could even be occurring at different levels, given the past literature. We have removed the speculation about different interneurons, instead saying that their different temporal characteristics may suggest distinct biologic mechanisms or local circuitry (page 21).

Methods: It may be useful to know how many animals data were collected at Hopkins vs Penn and if there were any differences. How many in each hemisphere by site?

We studied 2 animals at Hopkins and 3 at Penn, all had bilateral hemisphere data. We have noted this in the revised methods (page 22).

In line 424 is there a typo for the formula for RMI?

Thank you, very embarrassing for us. We have fixed it (page 25).

REVIEWER COMMENTS

Reviewer #1 (Remarks to the Author):

The authors have done an excellent job in revising their manuscript. This is a great paper!

Reviewer #2 (Remarks to the Author):

This reviewer appreciates the thorough additional data analyses that the authors have conducted in response to prior queries. The extensive extended data analysis and the reformatted figures are a significant positive change in this revision. These are exciting data.

However, where the paper really falls short in both setting up the problem and adequately describing the impact of the findings. Much is left to the reader to interpret the significance of these findings.

The introduction still lacks specific hypotheses which are well motivated and have strong rationale. This makes the analyses described in the results without structure. The discussion in turn also does not address clearly how the data in this paper can inform models of speech or vocal motor control, especially of vocal production with multiple phrases.

Other specific comments:

Abstract: Reference to distinct mechanisms and functions is vague. What does "these overlaps" refer to? How does the present approach exactly address this problem and disambiguate? The abstract is still too vague about the main finding and its implication.

What is distinct about the correlations with sensory tuning? Why does this necessarily mean that there are concurrent motor outputs based on recordings in auditory cortex? And how much of this is trivially due to the temporal characteristics of the vocalization itself?

Intro

Why should it be that "despite this prominent suppression ...". Tuning and sensitivity to feedback may reflect other computational processes. Unclear how this paper really addresses the origins or which underlying neural processes are unclear.

That suppression has both a generalized and a specific component has been shown by others - notably in the speaking induced suppression studies by Houde et al.

Furthermore, suppression and sensitivity to altered auditory feedback may also reflect distinct computational processes as has been shown by Chang et al.

What is the rationale for a "lower" vs "higher" corollary discharge signal? That seems to come from nowhere. At the end of line 101 a problem is identified by that approach towards solving it is not outlined clearly.

What is the evidence for a single global motor production plan in twitter vocalizations and for motor control for individual phrases?

Again the introduction is devoid of specific hypotheses with appropriate rationale and motivation that the paper seeks to test. Without this it is quite hard to evaluate the findings of the paper.

Figure 1 could include not only an individual twitter call but also the average of all the twitter calls. The PSTH responses are really aligned to the average call unless some clever time-warping procedure could be performed. It would be nice to include the raw traces of the same neuron during playback as well. Both peak suppressed units and excited are temporally aligned to the peak of the phrase onset - point that is not made here.

Additional the onset vs ongoing phrase analyses may be worth including in the main results, if motivated by specific hypotheses about differential mechanisms governing onset vs ongoing auditory stimulus processing during playback and during vocal production.

We thank the two reviewers for their additional comments on our revised manuscript. We have made changes to the manuscript, as described below, to address their comments and concerns. We feel that these revisions have made this a more cohesive and significant manuscript. Our changes in the manuscript have been colored in red, and their comments below have been italicized for clarity.

Reviewer #1 (Remarks to the Author):

The authors have done an excellent job in revising their manuscript. This is a great paper!

We thank reviewer for their enthusiasm.

Reviewer #2 (Remarks to the Author):

This reviewer appreciates the thorough additional data analyses that the authors have conducted in response to prior queries. The extensive extended data analysis and the reformatted figures are a significant positive change in this revision. These are exciting data.

We thank the reviewer for their enthusiasm.

However, where the paper really falls short in both setting up the problem and adequately describing the impact of the findings. Much is left to the reader to interpret the significance of these findings.

The introduction still lacks specific hypotheses which are well motivated and have strong rationale. This makes the analyses described in the results without structure. The discussion in turn also does not address clearly how the data in this paper can inform models of speech or vocal motor control, especially of vocal production with multiple phrases.

We thank the reviewer for their insightful comments. We have done extensive revisions of both the introduction and discussion in an attempt to better lay out the problem and interpretation. Notably, we have reframed the motivation a little, laying out the previous evidence for multiple vocal modulations, as suggested by Eddie Chang's seminal paper, and how our current paper attempts to reconcile why our past single unit data did not show this, and in fact showed a strong correlation of suppression and feedback sensitivity. We put forth the question about whether we could detect or disentangle these two processes, and show that they can both exist within individual neurons, rather than be a product of different groups of neurons (pgs 2-3). In our

discussion, we point the advance by this paper showing how these processes co-exist, but also target slightly different neural populations, with phasic/gating modulations being rather broadly distributed, and tonic/predictive modulations being more specific to vocal frequency range neurons, which may better support its role in self-monitoring. We have expanded our discussion about possible functions of these two processes, particularly tonic suppression, putting forward several possible functions in the context of self-monitoring (pg 22), but also discuss how multiple time scales might play a role in fluent speech, expanding on our previous brief discussion about segmental vs. suprasegmental vocal parameters (pg 22-23).

Other specific comments:

Abstract: Reference to distinct mechanisms and functions is vague. What does "these overlaps" refer to? How does the present approach exactly address this problem and disambiguate? The abstract is still too vague about the main finding and its implication.

What is distinct about the correlations with sensory tuning? Why does this necessarily mean that there are concurrent motor outputs based on recordings in auditory cortex? And how much of this is trivially due to the temporal characteristics of the vocalization itself?

We apologize about the vagueness and confusion. We have revised the abstract to clarify these issues in greater detail. We note overlaps both refer to our previously observed correlations of suppression and feedback as well as temporal overlaps of different input that prevent examining these individually. We have expanded the discussion of sensory tuning, noting that phasic suppression was broadly present regardless of units' CFs, while tonic was more specific and had greater feedback sensitivity. We would argue that these two processes with different properties beyond temporal characteristics are evidence of distinct motor inputs to AC, which we more fully discussed in the remainder of the manuscript.

Intro

Why should it be that "despite this prominent suppression ...". Tuning and sensitivity to feedback may reflect other computational processes. Unclear how this paper really addresses the origins or which underlying neural processes are unclear.

That suppression has both a generalized and a specific component has been shown by others - notably in the speaking induced suppression studies by Houde et al. Furthermore, suppression and sensitivity to altered auditory feedback may also reflect distinct computational processes as has been shown by Chang et al.

We have revised our introduction to be a clearer, noting the confusion arises from our past observations that feedback sensitive neurons were generally those that were suppressed, leading us to assume that they were a single process (pg 3). We have included a much more nuanced discussion of how this doesn't align with the human data (Chang et al), though other recordings in human Heschl's by the Iowa group have found a correlation of vocal and feedback responses (pg 3-4). We now point out that our objective is to try and reconcile our previous findings with the human observations, and determine whether we could find evidence of these two processes in single neurons, and whether they would both be present in individual neurons, or have distinct groups of neurons, as well as to determine if there was specificity in their targets (i.e. frequency tuning etc.) (pg 4-5).

What is the rationale for a "lower" vs "higher" corollary discharge signal? That seems to come from nowhere. At the end of line 101 a problem is identified by that approach towards solving it is not outlined clearly.

We chose to use the terminology of lower and higher corollary discharges as a way of conceptually grouping these signals that we borrowed from the literature (Crapse and Sommer 2008). Due to concerns previously raised about use of "Efferent Copy" vs. "Corollary Discharge" terminology, we sought an alternative way to group different notions of how corollary discharges affect sensory processing. The general notion is one of corollary discharges seen in the animal kingdom (both auditory and other sensory-motor systems) that are either gating or predictive. The former tend to be more temporally precise, which is part of the motivation to look at temporal patterns as a way to disentangle these processes. We have revised our introduction and removed the terms lower and higher, rather just referring to them as gating vs predictive corollary discharges (pg 4, 20-21).

What is the evidence for a single global motor production plan in twitter vocalizations and for motor control for individual phrases?

The reviewer brings up an important point, which we did not do an adequate job addressing previously. While it is generally accepted by the marmoset vocal community that a twitter is a single, cohesive vocal event and therefore likely involves a single motor plan, direct evidence is lacking. Previous work involving other marmoset calls (notably phee) has shown that the first phrase of multi-phrase calls predicts the acoustics of subsequent phrases, which has been interpreted as evidence of a global motor plan (Miller et al 2009), but similar evidence for twitters has not been published. We therefore performed additional analyses to show that, like phee, later twitter phrase acoustics are predicted by earlier phrases, which may be consistent with a more global plan beyond phrase specific control. This is now included in Extended Data Figure 1, discussed briefly in the results (pg 8-9), and a more extensive discussion added (pg 19-20).

Again the introduction is devoid of specific hypotheses with appropriate rationale and motivation that the paper seeks to test. Without this it is quite hard to evaluate the findings of the paper.

We thank the reviewer for their concerns, and hope that the revisions we have discussed above have addressed this concern.

Figure 1 could include not only an individual twitter call but also the average of all the twitter calls. The PSTH responses are really aligned to the average call unless some clever time-warping procedure could be performed. It would be nice to include the raw traces of the same neuron during playback as well. Both peak suppressed units and excited are temporally aligned to the peak of the phrase onset - point that is not made here.

While this is an excellent suggestion, unfortunately averaging all the twitters, whether in the time or spectrographic domain, results in an incomprehensible smeared mess. We had previously added histograms showing the average timing of twitter phrases to get a better sense of the phrase timing (Fig 1C, 1E). To give a reader a better sense of twitter variability, we have included spectrograms of 40 additional twitters in Extended Data Figure 1 as well as further analysis of twitter phrase and interval timing both for the unit shown in Fig 1 and for all our animals. We hope this adequately addresses the concern.

The PSTHs shown in Figure 1 are all aligned only to the onset of the twitter calls, and not an average call. We apologize for the confusion, and now specifically note this in the figure legends (pg 19). We had actually included a time-warped version of these PSTHs in our original manuscript submission, but it was suggested that we remove these as the reviewer did not feel they added much. We considered re-introducing them for this revision, but ultimately decided against it as the time warping introducing strange distortions due to the high degree of variability in phrase and interval durations.

As suggested, we have included raw raster plots for playback (Fig 1B).

As with the individual unit example, our suppressed and excited population PSTHs were actually aligned to twitter onset, rather than specific phrases. We had noted this in our original figure legend, but have revised legend to be more explicit (pg 19).

Additional the onset vs ongoing phrase analyses may be worth including in the main results, if motivated by specific hypotheses about differential mechanisms governing onset vs ongoing auditory stimulus processing during playback and during vocal production.

This is an excellent suggestion. We have performed additional analyses to compare first phrase responses to both ongoing phrases and intervals, as well as phasic and tonic activity. As in our earlier limited analysis there was a strong positive correlation, which we interpret as evidence that ongoing responses are not just some sort of adaptation process following a strong excitatory onset, which would be one alternative hypothesis to explain our results. A similar analysis was performed on playback, which showed very similar results. This new analysis is presented in Figure 5C and discussed in the manuscript (pg 15).

REVIEWERS' COMMENTS

Reviewer #2 (Remarks to the Author):

The authors are to be commended for the extraordinary amount of work put into the manuscript and for their responsiveness to prior critiques and for their diligence in incorporating these suggestions. This is now a vastly improved manuscript.

I only have a few minor suggestions for the abstract and the intro.

Abstract:

Also several studies have also shown that there are two components of predictions - the timing and the acoustics. The timing prediction is expected to be non-specific and the acoustic prediction is expected to show specificity or tuning. This has also been covered in previous work.

The behavioral role of these two processes remains unknown and so it is speculative to think that these phasic and tonic suppressive processes necessarily have different behavioral roles. Also what underlying mechanisms is being referred to here? Computational?

Intro Line 72, It is incorrect to posit that "it had been previously assumed that the two resulted from a single neural mechanism or process ...". In the field of motor control the suppression resulting from a comparison of expected auditory feedback with actual feedback is separate and distinct from a gain on the prediction errors when there are applied perturbations. Therefore, it is not accurate other than the authors own prior work.

The main hypotheses are still a bit unclear. Both gating and predictive processes are suppressive under normal feedback. This is not clearly stated in their hypotheses. I would recommend a rewriting to clarify that examine twitter calls allows us to look at multiple phrases of predictive processes - tonic and phasic and potentially disambiguate the timing and acoustic nature of the predictions.

We thank the editors for the chance to revise our manuscript, and the reviewers for their helpful comments. Below, we detail our response to the recent comments. We have italicized the comments for ease of reference.

The authors are to be commended for the extraordinary amount of work put into the manuscript and for their responsiveness to prior critiques and for their diligence in incorporating these suggestions. This is now a vastly improved manuscript.

I only have a few minor suggestions for the abstract and the intro.

We thank the reviewer for their insightful and pointed comments and suggestions over the many drafts of this manuscript. The final version is much more cohesive and better for their suggestions.

Abstract:

Also several studies have also shown that there are two components of predictions - the timing and the acoustics. The timing prediction is expected to be non-specific and the acoustic prediction is expected to show specificity or tuning. This has also been covered in previous work.

We thank the reviewer for their comment, we have revised the abstract to stress how we are trying to show these processes can occur for individual neurons rather than affecting different groups of neurons.

The behavioral role of these two processes remains unknown and so it is speculative to think that these phasic and tonic suppressive processes necessarily have different behavioral roles. Also what underlying mechanisms is being referred to here? Computational?

We thank the reviewer for the suggestions, while we speculate about possible behavioral roles in the discussion, we have removed it from the abstract. We have revised to also note we are referring to computational mechanisms (though we also speculate in the discussion about possible different circuit mechanisms)

Intro Line 72, It is incorrect to posit that "it had been previously assumed that the two resulted from a single neural mechanism or process ...". In the field of motor control the suppression resulting from a comparison of expected auditory feedback with actual feedback is separate and distinct from a gain on the prediction errors when there are applied perturbations. Therefore, it is not accurate other than the authors own prior work.

We thank the reviewer for the comments. There is actually some controversy in the speech field on this, there have been a few recent meeting posters suggesting a correlation between suppression and feedback prediction, as well as a manuscript our senior author has reviewed showing some evidence of this. Nonetheless, we have revised it to specifically say 'we have assumed' which is important, as thus far, the only published neuron-level data has come from our group.

The main hypotheses are still a bit unclear. Both gating and predictive processes are suppressive under normal feedback. This is not clearly stated in their hypotheses. I would recommend a rewriting to clarify that examine twitter calls allows us to look at multiple phrases of predictive processes - tonic and phasic and potentially disambiguate the timing and acoustic nature of the predictions.

We thank the reviewer for the suggestion, we have made several revision to be more specific about using twitters to disambiguate corollary discharge processes operating on multiple time-scales and their interactions with different acoustic inputs (page 5).